# Bimodal expression of Type 3 Secretion System 2 enables cooperative virulence among intracellular *Salmonella* Typhimurium

Milada Pospíšilová[1,2], Alona Dreus[1,2], Paulina Matheova[1,2], Jana Schmidtova[1], Barbora Pravdova[1], Michaela Blazikova[3], Martin Capek[3,4], Ondrej Cerny[1]*

1 Laboratory of Bacterial Virulence, Institute of Microbiology of the Czech Academy of Sciences, Prague, Czech Republic, 2 Faculty of Science, Charles University in Prague, Prague, Czech Republic, 3 Light Microscopy, Institute of Molecular Genetics of the Czech Academy of Sciences, Prague, Czech Republic, 4 Laboratory of Biomathematics, Institute of Physiology of the Czech Academy of Sciences, Prague, Czech Republic

* ondrej.cerny@biomed.cas.cz

## Abstract

The Type 3 Secretion System encoded by the *Salmonella* pathogenicity island 2 (SPI-2 T3SS) enables bacterial proliferation in *Salmonella*-containing vacuole and dissemination throughout the host. Despite its crucial importance, not all intracellular *Salmonella* express the SPI-2 T3SS. Using flow cytometry and microscopic analysis of bacteria within host cells, we demonstrate that both the injectisome as well as its effectors exhibit bimodal expression. This bimodality depended on activation of the transcriptional regulator SsrB by sensor kinase SsrA. Within an infected host cell, proliferation of bacteria not expressing the SPI-2 T3SS (SPI-2$^{OFF}$) depended on SPI-2 T3SS-expressing bacteria (SPI-2$^{ON}$), suggesting that all SPI-2 T3SS effectors necessary for *Salmonella* intracellular replication can be complemented *in trans*. SPI-2$^{OFF}$ bacteria had shorter division time *in vitro* and proliferated faster inside host cells than the SPI-2$^{ON}$ bacteria. SPI-2$^{OFF}$ bacteria egressed more from infected host cells, thus potentially serving as a reservoir for further *Salmonella* dissemination throughout the host. Collectively, our results suggest that bimodal expression of the SPI-2 T3SS and its effectors represents an adaptive mechanism that might increase *Salmonella* virulence.

## Author summary

Systemic infection and survival of *Salmonella* within host cells depend on expression of effector proteins that manipulate host cell functions. Here, we show that expression of these effectors varies among individual bacteria residing within the same host cell. Production of effectors by a subpopulation of intracellular *Salmonella* protects the non-producing bacteria and enables them to proliferate

**Data availability statement:** All data are in the manuscript and/or supporting information files and are also available on Zenodo (DOI: 10.5281/zenodo.17199173).

**Funding:** This work was supported by Czech Science Foundation (22-05356S to OC and 24-11259M to OC) and MSCA Fellowships CZ (CZ.02.01.01/00/22_010/0002357 to OC). We acknowledge support from Talking microbes - understanding microbial interactions within One Health framework (CZ.02.01.01/00/22_008/0004597 to OC). This work was supported by Grant Agency of Charles University (371025 to AD).The funders had no role in study design, data collection and analysis, decision to publish, or preparation of the manuscript.

**Competing interests:** The authors have declared that no competing interests exist.

efficiently inside the *Salmonella*-containing vacuole. This bimodal expression pattern is driven by activation of the master transcriptional regulator SsrB. *Salmonella* can modulate the bimodal pattern of effector expression and thus potentially optimize energetic burden across the bacterial population. Over time, the proportion of bacteria exhibiting lower effector expression increases within infected cells. Ultimately, the non-expressing bacteria egress more efficiently from the host cells and can therefore serve as a reservoir for further infection of bystander host cells.

## Introduction

To establish a replicative niche inside a host and spread to a new host eventually, bacterial pathogens must rapidly adapt to new environments and new threats. After entering a new host, bacteria encounter dramatic changes in temperature and pH, nutrient availability, competition with local microbiota, antibacterial peptides and host immune system [1]. Overcoming these challenges often requires expression of diverse gene clusters regulated through distinct and sometimes mutually exclusive ways. The evolutionary pressure for survival in the face of these diverse threats has likely driven the development of regulatory mechanisms that enable stochastic transcriptional activation or repression of particular genes, thus leading to the appearance of phenotypically distinct subpopulations. The formation of such subpopulations allows for bet-hedging as well as division of labour and cooperative virulence [2,3]. Among others, the model bacterial pathogen *Salmonella enterica* serovar Typhimurium has been extensively used to study the role of isogenic subpopulations in phenomena such as antibiotic persistence, transcriptional plasticity, intra- and extra-cellularity, and bimodal virulence factor expression [4–9].

*Salmonella* virulence depends primarily on type 3 secretion systems (injectisomes) encoded on the *Salmonella* pathogenicity islands 1 and 2 (SPI-1 T3SS and SPI-2 T3SS, respectively). The SPI-1 T3SS is essential for establishment of reproductive niche in host intestine, induction of local inflammation, invasion into nonphagocytic host cells and ultimately for transmission into a new host. Expression of the SPI-1 T3SS is regulated by complex signalling cascade in which the interaction between the repressor HilE and activator HilD dictates the final activity of the SPI-1 master regulator HilA. This regulation controls the formation of the SPI-1$^{ON}$ and SPI-1$^{OFF}$ subpopulations [10]. The SPI-1$^{ON}$ bacteria create the replicative niche in the host intestine, while the SPI-1$^{OFF}$ bacteria avoid the metabolic cost of the SPI-1 T3SS expression and can thus proliferate faster [11]. Surprisingly, cooperation between both subpopulations is crucial for efficient host cell invasion [8].

Upon entry into host cells, *Salmonella* survive within a specialised compartment termed *Salmonella*-containing vacuole (SCV). The SPI-2 T3SS is crucial for survival within the SCV and for bacterial dissemination throughout the host [12–14]. SPI-2 expression is induced by low nutrition availability and low pH within the SCV and is regulated mainly by the two-component regulatory system SsrA-SsrB. The sensor

kinase SsrA phosphorylates SsrB, enhancing its binding to the promoters in SPI-2 and those controlling expression of SPI-2 T3SS effectors [15]. SsrB binding displaces H-NS from these promoters and promotes recognition by RNA polymerase thereby activating transcription [16,17]. Although SPI-2 T3SS expression has been shown to be bimodal [18,19], the role of SsrA-SsrB interplay in regulation of this bimodality as well as the physiological significance of this bimodality remain unclear.

Here we show that not only the SPI-2 T3SS but also its effectors exhibit bimodal expression. The activating phosphorylation of SsrB by SsrA was crucial for establishing bimodal SPI-2 T3SS expression. This bimodality is not heritable and the subpopulation not expressing the SPI-2 T3SS effectors replenish the expressing subpopulation. Bacteria expressing SPI-2 T3SS support the proliferation of non-expressing bacteria within the SCV in the same host cell. Subsequently, the non-expressing bacteria then egress from the infected cells faster, potentially serving as a reservoir for infection of new host cells.

## Results

### Expression of SPI-2 T3SS is bimodal

To detect the expression of the SPI-2 T3SS in *Salmonella*, we examined the production of the SPI-2 T3SS needle protein SsaG using GFP reporter systems. We constructed the pP*ssaG*-GFP plasmid encoding GFP under the control of the *ssaG* promoter (P*ssaG*) in *Salmonella* strain encoding mCherry fluorescent protein on chromosome for easier identification of bacteria or used a *Salmonella* strain expressing GFP as a transcriptional fusion to *ssaG* (Fig 1A) to control for plasmid-specific artifacts.

To test the functionality of these reporters, we cultivated *Salmonella* P*ssaG* reporter strains in MgMES pH 5.0, a minimal medium mimicking the conditions in the SCV, for 5 h and monitored GFP fluorescence using fluorescence microscopy and flow cytometry. Approximately two thirds of bacteria from both reporter strains showed high GFP fluorescence suggesting that the GFP reporters monitoring P*ssaG* activity were functional *in vitro* (Fig 1B and 1C). As expected from a previous report [19], a subpopulation of bacteria did not show P*ssaG* activity during growth in MgMES pH 5.0 (Fig 1B and 1C).

To test the responsiveness of the P*ssaG* reporter, we compared the GFP fluorescence of *Salmonella* WT + pP*ssaG*-GFP under SPI-2-inducing and non-inducing conditions. No GFP fluorescence was detected in exponentially growing *Salmonella* in LB (Fig 1D), a condition inducing SPI-1 T3SS expression while repressing SPI-2 T3SS expression [20]. However, shifting bacteria to MgMES pH 5.0 minimal medium inducing SPI-2 expression resulted in strong GFP fluorescence in approximately 80% of *Salmonella* WT + pP*ssaG*-GFP. Statistical analysis [21] demonstrated that the distribution of bacteria into *ssaG*-non-expressing (SPI-2$^{OFF}$) and *ssaG*-expressing (SPI-2$^{ON}$) subpopulations followed a bimodal distribution.

Interestingly, *Salmonella* WT + pP*ssaG*-GFP grown to the late stationary phase in LB displayed a weak GFP fluorescent signal in approximately 25% of bacteria. However, these bacteria did not form a distinct population suggesting that they either express *ssaG* at low levels not comparable to the fully virulent state or that the detected GFP signal resulted from increased autofluorescence (Fig 1D).

To investigate the development of bimodal SPI-2 T3SS expression, we analysed the development of GFP fluorescence (*ssaG* expression) in MgMES pH 5.0 over time. The separation of GFP⁻ and GFP⁺ subpopulations was already apparent 0.5 h after the transfer of late stationary *Salmonella* WT + pP*ssaG*-GFP into MgMES pH 5.0 (Fig 1E), supporting an earlier finding that bacteria in stationary phase are 'SPI-2-primed' [20,22]. Although the vast majority of *Salmonella* WT + pP*ssaG*-GFP expressed GFP, a subpopulation remained GFP⁻ throughout the experiment (Fig 1E).

To examine the influence of *Salmonella* growth phase on bimodal SPI-2 T3SS expression, we also studied the development of *ssaG* expression in bacteria transferred to MgMES pH 5.0 from the late exponential growth phase (OD$_{600}$ = 1.8). As expected, the onset of GFP fluorescence in *Salmonella* WT + pP*ssaG*-GFP from the exponential culture was delayed

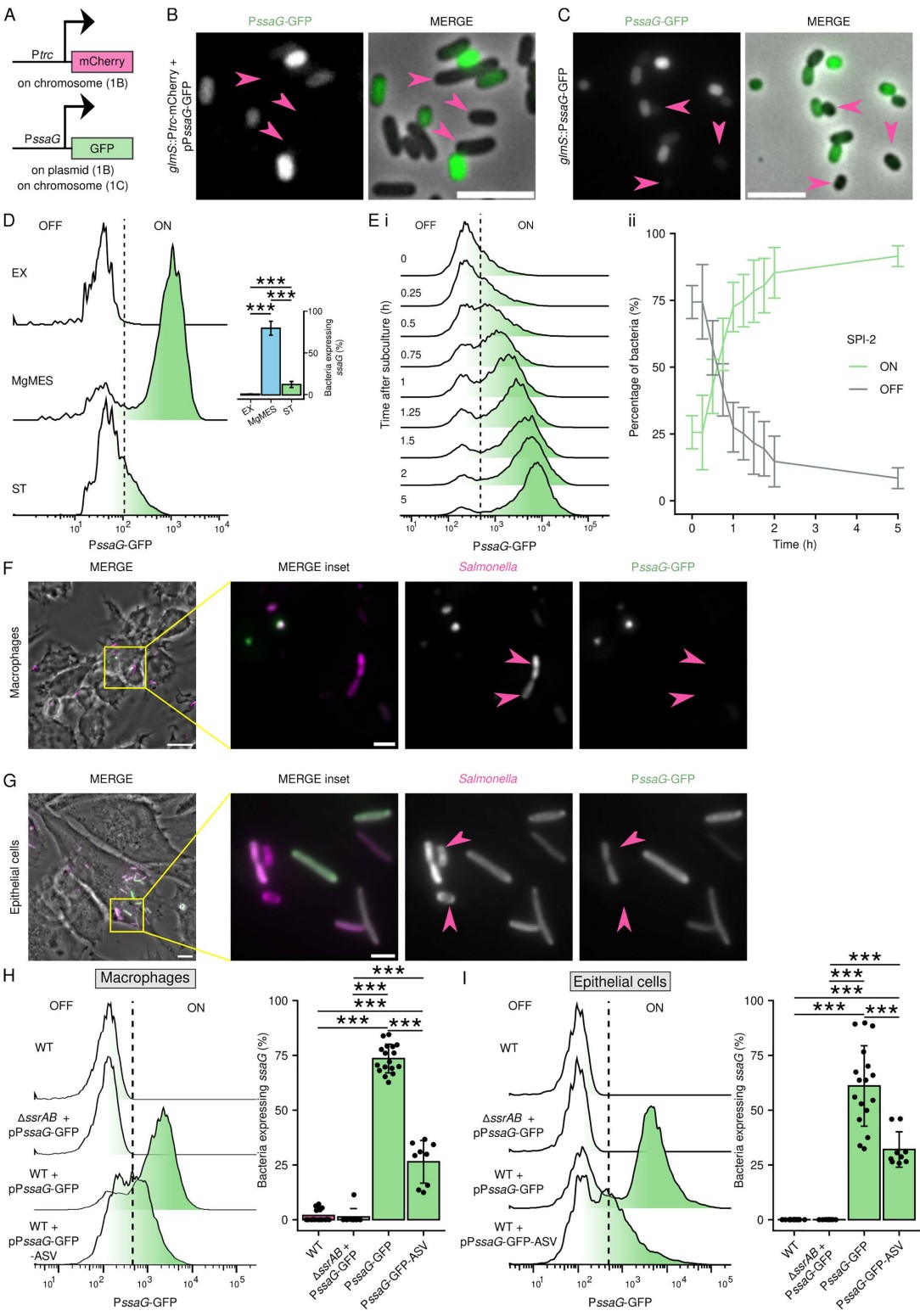

**Fig 1. Expression from SPI-2 T3SS is bimodal.** (A) Schematic of the fluorescent reporter system. The constitutive P*trc* regulated expression of mCherry from chromosome. GFP was under the control of the P*ssaG*, controlling expression of the SPI-2 T3SS needle protein, and was encoded either on a plasmid or on the chromosome (*ssaG* transcriptional fusion). (B, C) P*ssaG* activity *in vitro*. Representative fluorescence microscopy image of GFP expression

in STm WT + pPssaG-GFP (B) and STm ssaG::GFP (C) in a minimal medium MgMES pH 5 for 5 h. Scale bar, 2 µm. The arrowheads indicate bacteria with undetectable GFP fluorescence. (D) PssaG activity in SPI-2 inducing and non-inducing conditions. GFP expression in STm WT + pPssaG-GFP was analysed by flow cytometry at late exponential growth phase (OD 1.8 in LB, EX), in the minimal medium MgMES pH 5.0 (16 h, MgMES) and at late stationary growth phase (16 h in LB, ST). Negative gate (OFF) was set according to STm WT not carrying any reporter system. The bar chart shows quantification of GFP+ (ON) bacteria from 3 independent experiments in technical triplicates and show means ± SD. *** $p < 0.001$ (One-way ANOVA with Tukey HSD test). (E) PssaG activity over time. (i) GFP expression in STm WT + pPssaG-GFP was analysed by flow cytometry at indicated time after subculture from late stationary phase (16 h in LB) into MgMES pH 5. (ii) Quantification of GFP+ (ON) and GFP- (OFF) bacteria as represented in (i). Data are from 3 independent experiments in technical triplicates and show means ± SD. (F) PssaG activity in macrophages. Representative fluorescence microscopy image of GFP expression in STm WT + pPssaG-GFP in RAW264.7 16 h p.i. The arrowheads indicate bacteria with undetectable GFP fluorescence. Maximal z axis projection. Scale bar, 10 µm. (G) PssaG activity in epithelial cells. Representative fluorescence microscopy image of GFP expression in STm WT + pPssaG-GFP in Mel JuSo cells 24 h p.i. The arrowheads indicate bacteria with undetectable GFP fluorescence. Maximal z axis projection. Scale bar, 10 µm. (H) Bimodality of PssaG activity in macrophages. RAW264.7 macrophages were infected with STm WT, STm ΔssrAB + pPssaG-GFP, STm WT + pPssaG-GFP, or STm WT + pPssaG-GFP-ASV for 16 h before hypotonic lysis. PssaG activity was monitored as GFP fluorescence in bacteria from cell lysate by flow cytometry. The bar plot represents quantification of GFP+ (ON) bacteria. Data are from at least 3 independent experiments in technical triplicates and show means ± SD. ***$p < 0.001$ (Paired two-sample t-test). (I) Bimodality of PssaG activity in epithelial cells. Mel JuSo cells were infected with STm WT, STm ΔssrAB + pPssaG-GFP, STm WT + pPssaG-GFP, or STm WT + pPssaG-GFP-ASV for 24 h before hypotonic lysis. PssaG activity was monitored as GFP fluorescence in bacteria from cell lysate by flow cytometry. The bar plot represents quantification of GFP+ (ON) bacteria. Data are from at least 3 independent experiments in technical triplicates and show means ± SD. ***$p < 0.001$ (Paired two-sample t-test).

in comparison to bacteria from late stationary culture (S1A Fig). However, the proportion of GFP+ Salmonella was similar between bacteria transferred from the late exponential and stationary culture after 5 h (Figs 1E and S1A).

To analyse ssaG expression in cellulo, we infected RAW264.7 macrophages or Mel JuSo epithelial cells with Salmonella WT + pPssaG-GFP. Salmonella grown to the stationary phase is commonly used to infect macrophages, whereas Salmonella from the late exponential phase is used to infect epithelial cells as the bacteria express SPI-1, enabling host cell invasion by Salmonella. Both SPI-2OFF and SPI-2ON subpopulations were detected in macrophages (Fig 1F and 1H) as well as in epithelial cells (Fig 1G and 1I). Importantly, Salmonella lacking the ssrAB two-component system regulating SPI-2 expression did not show GFP fluorescence suggesting that the reporter expression is fully dependent on the described SPI-2 master regulator.

Since GFP molecules have stability longer than the duration of our experiments, we used a destabilised GFP version (GFP-ASV) to identify Salmonella that expressed the SPI-2 T3SS just before the end of the experiment [23]. Although the bimodal distribution was preserved, we observed fewer GFP+ bacteria suggesting a dynamic regulation of SPI-2 expression (Fig 1H and 1I).

Ratiometric analysis of GFP expression from PssaG and mCherry expression from a synthetic promoter suggests that the bimodal distribution of ssaG expression is not caused by factors influencing overall transcriptome (S1B and S1C Fig). Similarly, the presence of the PssaG reporter system does not represent a growth disadvantage that could impose a pressure for decreased GFP expression (S1D and S1E Fig). Collectively, these data demonstrate that expression of the SPI-2 T3SS in host cells follows a bimodal pattern.

## Expression of SPI-2 T3SS effectors is also bimodal

To examine whether the expression of SPI-2 T3SS effectors follows a bimodal pattern similar to that of the SPI-2 T3SS, we constructed the reporter plasmid pPsifA-BFP_PsteD-GFP encoding mTagBFP2 (referred to as BFP hereinafter) under the control of the PsifA and GFP under the control of the PsteD (Fig 2A). While SifA is crucial for SCV stability and Salmonella growth in macrophages, SteD reduces the interaction between dendritic cells and T cells, inhibits antigen presentation [24–28] but is dispensable for Salmonella intracellular growth.

Simultaneous microscopic and flow cytometry analysis of PsifA and PsteD reporter activity in Salmonella WT + pPsifA-BFP_PsteD-GFP grown in MgMES pH 5.0 (Fig 2B) as well as in bacteria inside host cells (Fig 2C–2E) revealed heterogeneous expression of the fluorescent reporter proteins. We detected bacteria not expressing either of the studied effectors, bacteria expressing higher amounts of sifA than steD, and bacteria expressing similar or higher amounts of steD than

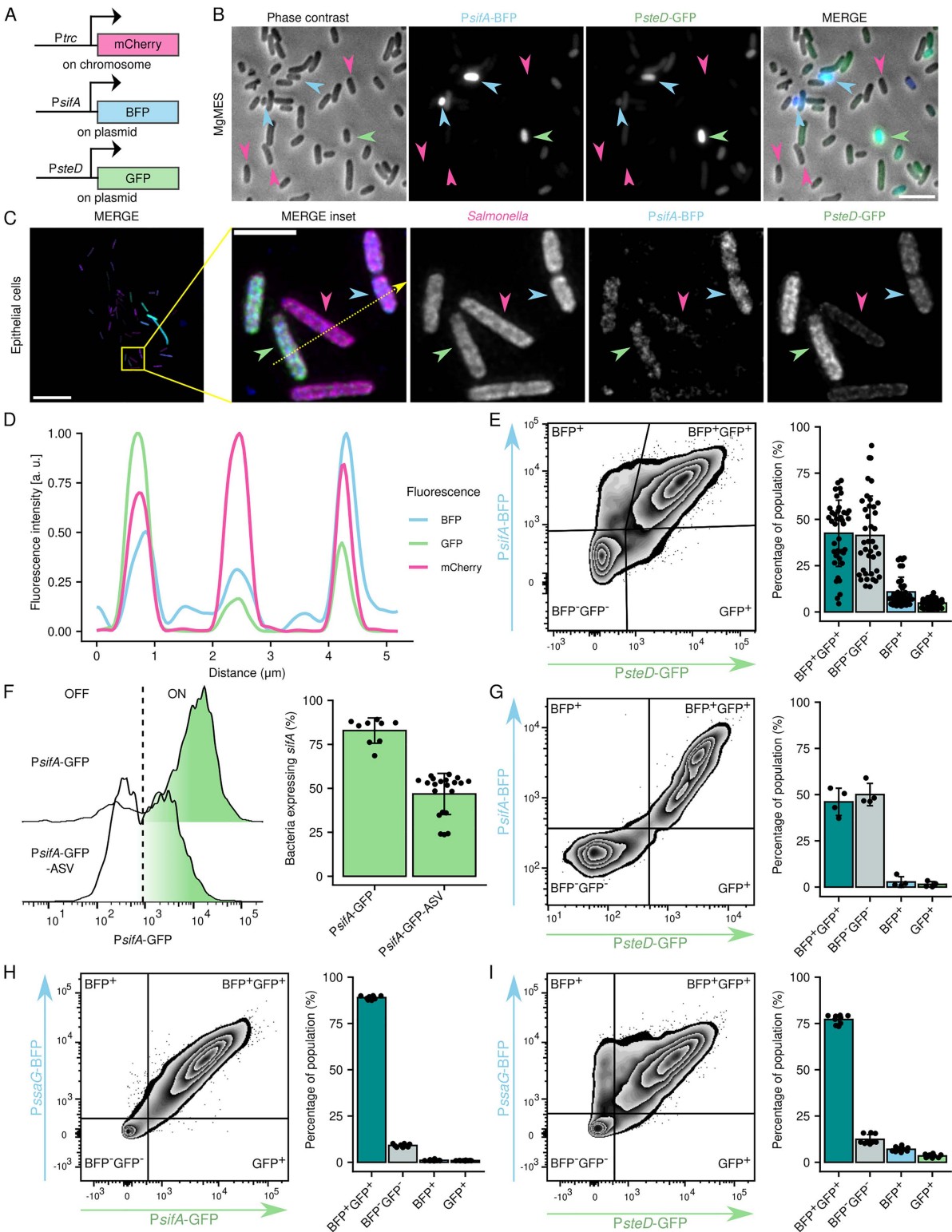

**Fig 2. Expression of SPI-2 T3SS effectors is also bimodal.** (A) Schematic of the fluorescent reporter system. The constitutive P*trc* regulated expression of mCherry from chromosome. BFP and GFP were under the control of the *sifA,* and *steD* promoter on a plasmid, respectively. (B) Activity of P*sifA* and P*steD* expression *in vitro*. Representative fluorescence microscopy image of BFP and GFP expression in *S*Tm WT + pP*sifA*-BFP_P*steD*-GFP in MgMES pH 5 for 5 h. The magenta arrow heads indicate expressing neither BFP nor GFP, blue arrowheads point to bacteria expressing more BFP than

GFP, and green arrowhead points to a bacterium expressing more GFP than BFP. Scale bar, 2 μm. (C) Activity of P*sifA* and P*steD* in epithelial cells. Representative fluorescence microscopy image of BFP and GFP expression in *S*Tm WT + pP*sifA*-BFP_P*steD*-GFP in Mel JuSo cells 24 h p.i. Magenta arrowhead points to a bacterium expressing neither BFP nor GFP, blue arrowhead points to a bacterium expressing more BFP than GFP, and green arrowhead points to a bacterium expressing more GFP than BFP. Maximal Z axis projection. Scale bar, 10 μm. (D) Fluorescence profile of mCherry (*Salmonella*), BFP (P*sifA*), and GFP (P*steD*) signal on the dotted line shown in (*C*). (E) Bimodality of P*sifA* and P*steD* activity in macrophages. RAW264.7 macrophages were infected with *S*Tm WT + pP*sifA*-BFP_P*steD*-GFP for 16 h before hypotonic lysis. P*sifA* and P*steD* activity was monitored as BFP and GFP fluorescence, respectively, in bacteria from cell lysate by flow cytometry. The bar plot represents quantification of bacteria in depicted gates. Data are from 22 independent experiments in technical triplicates and show means ± SD. (F) Bimodality of P*sifA* activity measured using destabilized reporter in macrophages. RAW264.7 macrophages were infected with *S*Tm WT + pP*sifA*-GFP, or *S*Tm WT + pP*sifA*-GFP-ASV for 16 h before hypotonic lysis. P*sifA* activity was monitored as GFP fluorescence in bacteria from cell lysate by flow cytometry. The bar plot represents quantification of bacteria in depicted gates. Data are from 7 independent experiments in technical triplicates and show means ± SD. (G) Bimodality of P*sifA* and P*steD* activity *in vivo*. C57BL/6 mice were inoculated orally with *S*Tm WT + pP*sifA*-BFP_P*steD*-GFP. 5 days post inoculation, individual splenocytes were isolated and lysed as in (*D*). P*sifA* and P*steD* activity was monitored as BFP and GFP fluorescence, respectively, in bacteria from cell lysate by flow cytometry. The bar plot represents quantification of bacteria in individual quadrants. Data are from 4 independently injected mice and show means ± SD. (H) Bimodality of P*ssaG* and P*sifA* activity in macrophages. RAW264.7 macrophages were infected with *S*Tm WT + pP*ssaG*-BFP_P*sifA*-GFP for 16 h before hypotonic lysis. P*ssaG* and P*sifA* activity was monitored as BFP and GFP fluorescence, respectively, in bacteria from cell lysate by flow cytometry. The bar plot represents quantification of bacteria in depicted gates. Data are from 3 independent experiments in technical triplicates and show means ± SD. (I) Bimodality of P*ssaG* and P*steD* activity in macrophages. RAW264.7 macrophages were infected with *S*Tm WT + pP*ssaG*-BFP_P*steD*-GFP for 16 h before hypotonic lysis. P*ssaG* and P*steD* activity was monitored as BFP and GFP fluorescence, respectively, in bacteria from cell lysate by flow cytometry. The bar plot represents quantification of bacteria in depicted gates. Data are from 3 independent experiments in technical triplicates and show means ± SD.

*sifA*. Similarly to SPI-2 expression, the expression of individual effectors also showed dynamic pattern as the destabilized pP*sifA*-GFP-ASV reporter (Figs 2F and S2A) also showed bimodal GFP distribution but overall less GFP⁺ bacteria. These results suggest that not all bacteria express all effectors in identical pattern.

Examination of reporter fluorescence kinetics upon bacterial transfer to MgMES pH 5.0 from late exponential or stationary phase revealed a bimodal expressional pattern for both P*sifA* (S1G and S1I Fig) and P*steD* (S1H and S1J Fig) that was similar to that of the P*ssaG* reporter. Importantly, bimodal expression from P*sifA* and P*steD* was also recapitulated in spleens of C57BL/6 mice 5 days after oral infection with *Salmonella* WT + pP*sifA*-BFP_P*steD*-GFP (Fig 2G).

To compare the expression of *sifA* and *steD* with that of the SPI-2 T3SS, we constructed the reporter plasmids pP*ssaG*-BFP_P*sifA*-GFP and pP*ssaG*-BFP_P*steD*-GFP. Infection of RAW264.7 macrophages showed that all SPI-2^ON bacteria also expressed *sifA* (Fig 2H). Interestingly, a subpopulation of the SPI-2^ON bacteria did not express *steD* (BFP⁺; Fig 2I).

To examine expression differences between effectors linked to *Salmonella* proliferation in the SCV and other SPI-2 T3SS effectors, we compared expression patterns of *sifA* and *steD* with *sseJ*, involved in SCV stability [29], and *steC*, influencing actin cytoskeleton remodelling and cell migration [30,31]. To limit the potential impact of bacterial growth phase on the reporter system, we infected epithelial Mel JuSo cells with bacteria in late exponential phase ($OD_{600}$ = 1.8). Similar to *Salmonella* WT + pP*sifA*-BFP_P*steD*-GFP (S2B Fig), *Salmonella* WT + pP*sseJ*-BFP_P*steD*-GFP formed a subpopulation expressing *sseJ*, but not *steD* (BFP⁺; S2C Fig). Infection of Mel JuSo cells with *Salmonella* WT + pP*sifA*-BFP_P*sseJ*-GFP led to formation of only BFP⁻GFP⁻ and BFP⁺GFP⁺ subpopulations (S2D Fig).

Infection of Mel JuSo cells with *Salmonella* WT + pP*steC*-BFP_P*sifA*-GFP showed that expression of *steC* followed a pattern similar to *sifA* and *sseJ*. Infection of Mel JuSo cells with *Salmonella* WT + pP*steC*-BFP_P*sifA*-GFP resulted in formation of only BFP⁻GFP⁻ and BFP⁺GFP⁺ subpopulations (S2E Fig). In contrast, infection of Mel JuSo cells with *Salmonella* WT + pP*steC*-BFP_P*steD*-GFP resulted in formation of small BFP⁺ bacterial subpopulation expressing *steC* but not *steD* in addition to bacteria expressing both or neither effector (S2F Fig). Furthermore, we examined expression of several other effectors, all showing bimodal expressional pattern (S2G Fig).

To further examine the subpopulation expressing only one effector, we compared the BFP⁺ subpopulations among individual reporter *Salmonella* strains. We detected significantly more bacteria expressing *sifA* and *sseJ* than *steC*, while *steD* was expressed by the smallest bacterial proportion (S2H and S2I Fig). Considering that SifA and SseJ are involved in SCV function, SteC influences functions of the infected cell, while SteD shows paracellular activity, we hypothesise that

the percentage of bacteria expressing particular effectors might depend on the importance of the effector for intracellular bacterial survival (i.e., SCV function > cytosolic function > paracellular function; S2H and S2I Fig).

Collectively, these data show that the bimodal expression of individual effectors results in the formation of distinct *Salmonella* subpopulations. We have also observed that effectors more critical for *Salmonella* survival being expressed by a greater proportion of bacteria. Thus, careful examination of effector expressional pattern might help with uncovering its physiological function.

## SPI-2$^{OFF}$ phenotype is not heritable

To test if the formation of individual subpopulations was caused by a heritable mutation, we infected Mel JuSo cells with *Salmonella* WT + pP*sifA*-BFP_P*steD*-GFP, sorted out the BFP$^-$GFP$^-$, BFP$^+$, and BFP$^+$GFP$^+$ bacteria, and examined their capacity to reestablish these populations after reinfection of Mel JuSo cells (Fig 3A). As expected, the first infection resulted in formation of all three subpopulations (Fig 3B). After regrowing the sorted bacteria to identical physiological state in the late exponential phase (OD$_{600}$ = 1.8) and reinfecting Mel JuSo cells, the bacteria from all three subpopulations again formed all three subpopulations in unchanged proportions (Fig 3C). This shows that no heritable defect was responsible for bimodal effector expression.

## Bimodal expression of SPI-2 T3SS and its effectors depends on mode of SsrB activation

To identify the mechanism regulating the bimodality of expression of the SPI-2 T3SS and its effectors, we first examined viability and metabolic capacity of SPI-2$^{OFF}$ *Salmonella.* We constructed the reporter plasmid pP*ssaG*-BFP_P*BAD*-GFP encoding BFP under the control of the P*ssaG*, allowing the detection of SPI-2$^{ON}$ bacteria, and GFP under the control of the inducible arabinose promoter *BAD*. Incubation of RAW264.7 macrophages infected with *Salmonella* WT + pP*ssaG*-BFP_P*BAD*-GFP with arabinose for 4 h led to GFP expression in approximately 90% of BFP$^+$ bacteria (S3A Fig), suggesting that not all living bacteria (actively expressing BFP) respond to arabinose. Among BFP$^-$ *Salmonella*, about 60% responded to arabinose, confirming that while some of BFP$^-$ bacteria might be dead, the majority of SPI-2$^{OFF}$ *Salmonella* were alive and sharing some metabolic capabilities with SPI-2$^{ON}$ *Salmonella* inside host cells (S3A Fig). Similarly, the SPI-2$^{OFF}$ and SPI-2$^{ON}$ subpopulations showed no differences in the mCherry expression from the constitutive P*trc* (S1F Fig), suggesting that the global expression pattern did not differ between these subpopulations in RAW264.7 macrophages.

Following the invasion of epithelial cells, a fraction of *Salmonella* escape from the SCV, where the SPI-2 expression could be induced [32,33]. To determine whether SPI-2$^{OFF}$ *Salmonella* reside in the SCV or are present in the host cytosol, we ectopically expressed the SCV marker LAMP1-mTurquoise in Mel JuSo cells before infecting them with *Salmonella* WT + pP*ssaG*-GFP. We detected no difference in the localisation of LAMP1-mTurquoise to the vicinity of GFP$^+$ and GFP$^-$ bacteria (S3B Fig), suggesting that SPI-2$^{OFF}$ bacteria reside in the SCV.

To investigate the involvement of signalling events and transcriptional regulators responding to the SCV environment, we constructed *Salmonella* Δ*ssrAB* + pP*ssaG*-GFP lacking SsrB, the master transcriptional regulator of SPI-2 T3SS expression, and its activator SsrA. As expected, *Salmonella* Δ*ssrAB* + pP*ssaG*-GFP failed to express GFP in response to nutrient limitation and acidic conditions in MgMES pH 5.0 (Fig 4A). In contrast, constitutive overexpression of SsrB$^{D56E}$-2HA (S4A Fig) with a point phosphomimetic mutation that enhances SsrB activity by mimicking the SsrB activation by SsrA inside the SCV [34] led to constitutive GFP expression even in SPI-2-repressing conditions during exponential growth (Figs 4A and S4B). Importantly, the GFP$^-$ subpopulation was absent in *Salmonella* Δ*ssrAB* + pP*trc*-*ssrB*$^{D56E}$ + pP*ssaG*-GFP under all tested conditions, suggesting that either the presence or activity of SsrB is involved in regulation of bimodality of SPI-2 T3SS expression.

To determine whether the bimodality of SPI-2 T3SS expression is regulated by the presence or activity of SsrB, we infected RAW264.7 macrophages with *Salmonella* WT + pP*ssaG*-GFP, *Salmonella* Δ*ssrAB* + pP*ssaG*-GFP, *Salmonella* Δ*ssrAB* + pP*trc*-*ssrB*$^{WT}$ + pP*ssaG*-GFP, and *Salmonella* Δ*ssrAB* + pP*trc*-*ssrB*$^{D56E}$ + pP*ssaG*-GFP. As expected, flow cytometry

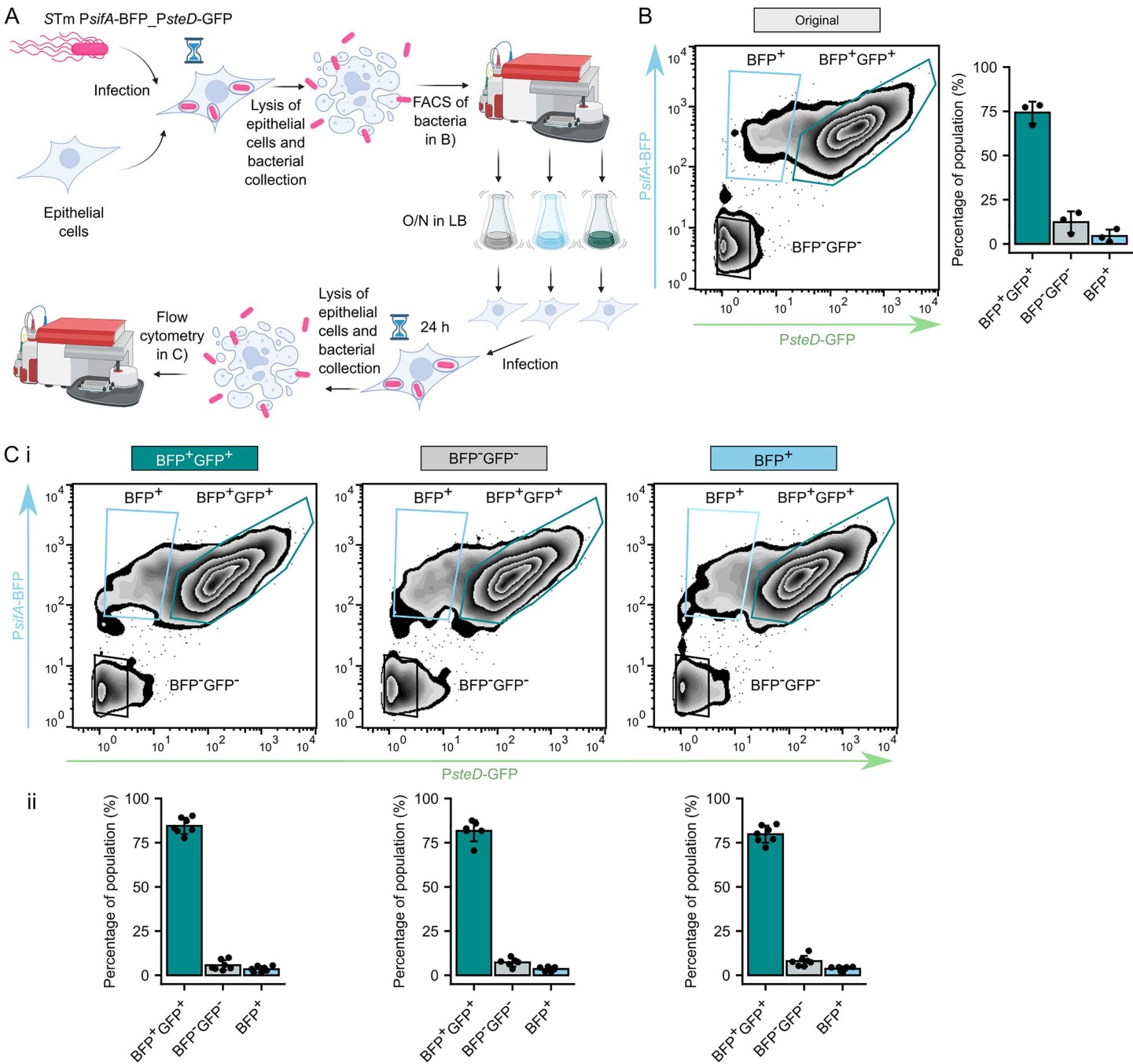

**Fig 3. SPI-2$^{OFF}$ phenotype is not heritable.** (A) Schematic showing experimental procedure to track heritability of bimodal effector expression. Mel JuSo cells were infected with STm WT + pP*sifA*-BFP_P*steD*-GFP for 24 h before hypotonic lysis. Liberated bacteria were FACS sorted based on their fluorescence intensity of BFP and GFP into BFP⁻GFP⁻, BFP⁺, and BFP⁺GFP⁺ populations. Sorted bacteria were grown separately in LB medium and then used to reinfect freshly prepared Mel JuSo cells for 24 h before hypotonic lysis. P*sifA* and P*steD* activity was monitored as BFP and GFP fluorescence, respectively, in bacteria from cell lysate by the instrument used for FACS. Created in BioRender. Pospíšilová, M. (2025) https://BioRender.com/yzlr0ea. (B) Data obtained during the original sorting. Data are from 3 independent experiments in technical triplicates and show means ± SD. (C) Data obtained from reinfection of Mel JuSo cells with bacteria from the BFP⁻GFP⁻, BFP⁺, and BFP⁺GFP⁺ bacterial populations. (i) Representative zebra plots showing bacterial populations in reinfection samples. (ii) Quantification. Data are from 3 independent experiments in technical triplicates and show means ± SD.

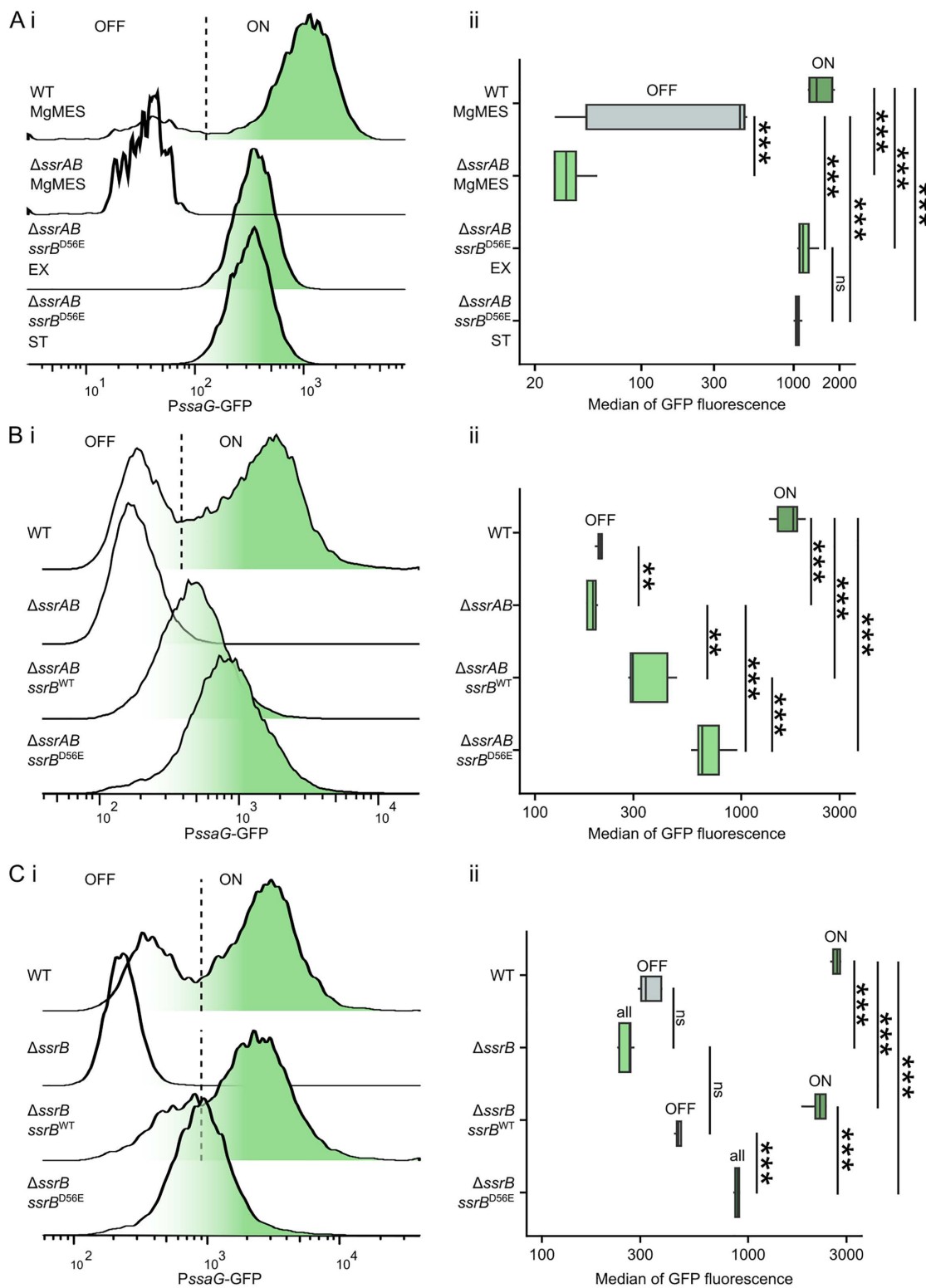

**Fig 4. Bimodal expression of SPI-2 T3SS and its effectors depends on activation of SsrB.** (A) P*ssaG* activity *in vitro*. (*i*) P*ssaG* activity was monitored as GFP fluorescence in STm WT+pP*ssaG*-GFP, STm Δ*ssrAB*+pP*ssaG*-GFP, and STm Δ*ssrAB*+P*trc*-*ssrB*^D56E^+pP*ssaG*-GFP by flow cytometry after 16 h of growth in MgMES pH 5, at the late exponential phase OD 1.8 (EX) in LB (when SPI-1 expression is induced), or at the late stationary

phase (16 h) in LB (when SPI-2 expression is primed). P*ssaG* activity was monitored as GFP fluorescence in bacteria by flow cytometry. (*ii*) Quantification of GFP MFI in bacteria as represented in (*i*). Data are from 3 independent experiments in technical triplicates and show means ± SD. ns, p > 0.05; ***p < 0.001 (One-way ANOVA with Tukey HSD test). (B) Influence of SsrB on modality of P*ssaG* activity in macrophages. (*i*) RAW264.7 macrophages were infected with *S*Tm WT + pP*ssaG*-GFP, *S*Tm Δ*ssrAB* + pP*ssaG*-GFP, Δ*ssrAB* + P*trc*-*ssrB*^WT + pP*ssaG*-GFP, or *S*Tm Δ*ssrAB* + P*trc*-*ssrB*^D56E + pP*ssaG*-GFP for 16 h prior hypotonic lysis. P*ssaG* activity was monitored as GFP fluorescence in bacteria from cell lysate by flow cytometry. (*ii*) Quantification of GFP MFI in bacteria as represented in (*i*). Data are from 3 independent experiments in technical triplicates and show means ± SD. **p > 0.01; ***p < 0.001 (One-way ANOVA with Tukey HSD test). (C) Influence of SsrB on modality of P*ssaG* activity in presence of SsrA. (*i*) RAW264.7 macrophages were infected with *S*Tm WT + pP*ssaG*-GFP, *S*Tm Δ*ssrB* + pP*ssaG*-GFP, Δ*ssrB* + P*trc*-*ssrB*^WT + pP*ssaG*-GFP, or *S*Tm Δ*ssrB* + P*trc*-*ssrB*^D56E + pP*ssaG*-GFP for 16 h prior hypotonic lysis. P*ssaG* activity was monitored as GFP fluorescence in bacteria from cell lysate by flow cytometry. (*ii*) Quantification of GFP MFI in bacteria as represented in (*i*). Data are from 3 independent experiments in technical triplicates and show means ± SD. ns, p > 0.05; ***p < 0.001 (One-way ANOVA with Tukey HSD test).

analysis of P*ssaG* activity in intracellular bacteria showed that macrophages infected with *Salmonella* WT + pP*ssaG*-GFP contained SPI-2^ON and SPI-2^OFF bacteria, whereas macrophages infected with *Salmonella* Δ*ssrAB* + pP*ssaG*-GFP contained only the SPI-2^OFF bacteria (Fig 4B). Importantly, constitutive overexpression of *ssrB*^WT in *Salmonella* Δ*ssrAB* + pP*trc*-*ssrB*^WT + pP*ssaG*-GFP resulted in weak, unimodal P*ssaG* activity in the absence of *ssrA*. In contrast, constitutive overexpression of SsrB^D56E in *Salmonella* Δ*ssrAB* + pP*trc*-*ssrB*^D56E + pP*ssaG*-GFP significantly increased P*ssaG* activity yet more, without affecting its unimodal distribution (Figs 4B and S4C).

To dissect the individual roles of SsrA and SsrB in regulation of the bimodality of the SPI-2 T3SS expression, we infected RAW264.7 macrophages with *Salmonella* WT + pP*ssaG*-GFP, *Salmonella* Δ*ssrB* + pP*ssaG*-GFP with intact *ssrA* but deleted *ssrB*, and its complemented variants *Salmonella* Δ*ssrB* + pP*trc*-*ssrB*^WT + pP*ssaG*-GFP and *Salmonella* Δ*ssrB* + pP*trc*-*ssrB*^D56E + pP*ssaG*-GFP. As expected, *ssrA* alone was not sufficient for induction of detectable P*ssaG* activity (Fig 4C). Constitutive overexpression of *ssrB*^WT in *Salmonella* Δ*ssrB* + pP*trc*-*ssrB*^WT + pP*ssaG*-GFP restored bimodal P*ssaG* activity. The P*ssaG* activity in the SPI-2^ON subpopulation of *Salmonella* Δ*ssrB* + pP*trc*-*ssrB*^WT + pP*ssaG*-GFP almost matched that of the SPI-2^ON subpopulation in *Salmonella* WT + pP*ssaG*-GFP. In contrast, constitutive overexpression of SsrB^D56E in *Salmonella* Δ*ssrB* + pP*trc*-*ssrB*^D56E + pP*ssaG*-GFP led to unimodal P*ssaG* activity that was, however, lower than that in SPI-2^ON *Salmonella* Δ*ssrB* + pP*trc*-*ssrB*^WT + pP*ssaG*-GFP (Fig 4C). Collectively, these results suggest that while SsrB is essential for expression of the SPI-2 T3SS, the bimodality is regulated at the level of SsrB activation by SsrA rather than at the level of SsrB expression.

## SPI-2^OFF bacteria benefit from activity of effectors translocated by SPI-2^ON bacteria in macrophages

To analyse the benefits of bimodal expression of the SPI-2 T3SS and its effectors inside host cells, we examined the growth benefit of SPI-2^OFF *Salmonella* in the presence or absence of SPI-2^ON bacteria. To mimic the SPI-2^OFF population, we used the *Salmonella* ΔSPI-2 strain lacking the entire SPI-2 island including the *ssrAB* two-component system. *Salmonella* ΔSPI-2 strains are deficient in survival and proliferation in macrophages [35–37]. As a source of SPI-2^ON bacteria, we used *Salmonella* WT, since the phosphomimetic *ssrB*^D56E inducing the SPI-2^ON bacteria also activates the expression of genes beyond those encoding the SPI-2 T3SS and its effectors [38,39].

First, we injected C57BL/6 mice intraperitoneally with either a 1:1 mixture of *Salmonella* mCherry WT and *Salmonella* GFP ΔSPI-2 or with each strain individually. The presence of bacteria in splenic macrophages was determined by flow cytometry 24 h p.i. (S5A Fig). As expected, we recovered significantly fewer *Salmonella* GFP ΔSPI-2-infected macrophages than *Salmonella* mCherry WT-infected macrophages from the spleens of mice injected with individual bacterial strains (S5B Fig). However, *Salmonella* GFP ΔSPI-2 exhibited significant growth benefit in macrophages that also contained *Salmonella* mCherry WT compared to *Salmonella* GFP ΔSPI-2-infected macrophages from the same double-infected mouse. (Figs 5A, 5B, S5C and S5D). However, this experimental setup does not distinguish between bacteria phagocytosed and transported to the spleen at different times p.i. and thus cannot control for import of bacteria proliferating under different selection pressure in other tissues.

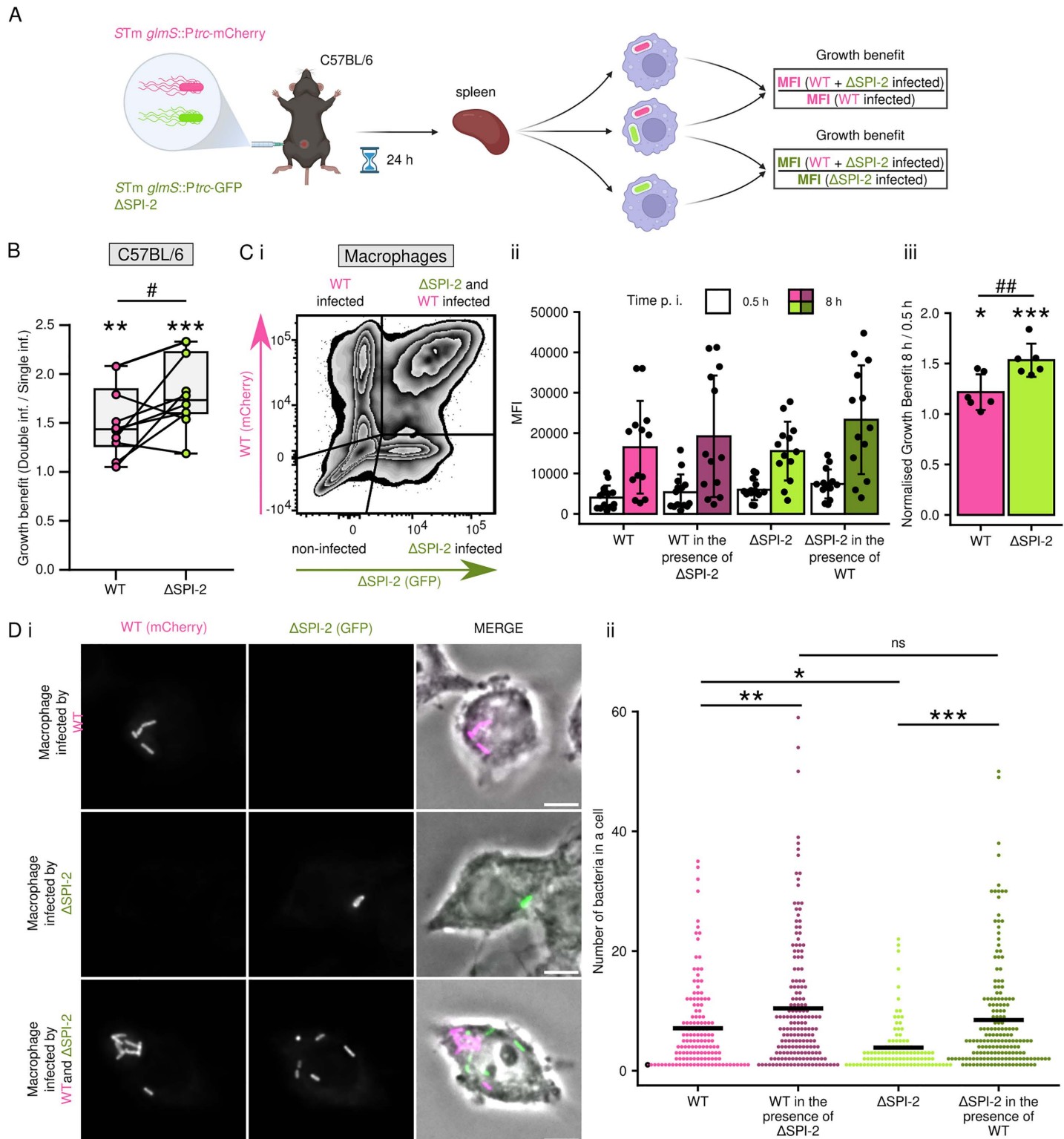

**Fig 5. SPI-2^OFF bacteria benefit from activity of effectors translocated by SPI-2^ON bacteria in macrophages.** (A) Schematic of mouse coinfection simulating SPI-2^ON and SPI-2^OFF populations. STm glmS::GFP ΔSPI-2 simulating the SPI-2^OFF population and STm glmS::mCherry WT providing the SPI-2^ON population were mixed 1:1 and injected intraperitoneally into C57BL/6. 24 h post injection, spleens were dissociated and STm WT and STm glmS

ΔSPI-2 fluorescence was examined in splenic macrophages by flow cytometry. The ratio of MFI of a particular fluorescent protein in double- and single-infected splenic macrophages was calculated as the growth benefit. Created in BioRender. Pospíšilová, M. (2025) https://BioRender.com/zngh9p8. (B) Presence of STm WT supports growth of STm ΔSPI-2 in vivo. C57BL/6 mice were injected intraperitoneally as described in A). The growth benefit values for using STm glmS::mCherry WT and STm glmS::GFP ΔSPI-2 from the same mouse are linked. Data are from 3 independent experiments in technical triplicates and show medians, Q1 and Q3. *p < 0.05, **p < 0.01, ***p < 0.001 (One-sample t-test, where hypothesized population mean (μ) = 1; # p < 0.05 paired two-sample t-test). (C, D) STm ΔSPI-2 has a growth benefit in the presence of STm WT in macrophages. RAW264.7 macrophages were infected with 1:1 mixture of STm glmS::mCherry WT and STm glmS::GFP ΔSPI-2. The amount of bacteria per one macrophage was determined at 0.5 and 8 h p.i. using and flow cytometry (C) and fluorescence microscopy (D). (C) (i) Representative flow cytometry zebra plot. (ii) Median fluorescence intensities (MFI) of populations 0.5 h p.i. (white columns) and 8 h p.i. (coloured columns). (iii) The growth benefit is calculated as the ratio of MFI of double-infected to MFI of single-infected splenic macrophages at 8 h p.i. normalized to the corresponding ratio at 0.5 h p.i. (also described in methods). Data are from 3 independent experiments in technical triplicates and show individual data points and means ± SD. *p < 0.05, ***p < 0.001 (One-sample t-test, where hypothesized population mean (μ) = 1; ## p < 0.01 paired two-sample t-test). (D) (i) Representative fluorescence microscopy images. Scale bars, 10 μm. (ii) Quantification of intracellular bacteria 8 h p.i. from (i). Data are from 3 independent experiments in technical triplicates and show individual data points and means. ns, *p < 0.05, **p < 0.01, ***p < 0.001 (Two-way ANOVA with Tukey HSD test).

To conduct the experiment in a synchronised manner, we repeated this experiment *in cellulo* using RAW264.7 macrophages. As expected, *Salmonella* GFP ΔSPI-2 proliferated less than *Salmonella* mCherry WT in RAW264.7 macrophages infected with individual strains (S5F Fig). Flow cytometry analysis of bacterial numbers in RAW264.7 cells showed that *Salmonella* GFP ΔSPI-2 had a significant growth benefit in macrophages that also contained *Salmonella* mCherry WT when compared to *Salmonella* GFP ΔSPI-2 in macrophages from the same sample but infected with the mutant alone (Figs 5C and S5E). A small, but significant growth benefit was also observed for *Salmonella* mCherry WT in double-infected macrophages, although this growth benefit was significantly smaller than that of *Salmonella* GFP ΔSPI-2 in the same macrophages (Fig 5C). Fluorescence microscopy revealed that *Salmonella* GFP ΔSPI-2 proliferated to the same levels as *Salmonella* mCherry WT in co-infected macrophages, whereas it exhibited a growth defect in single-infected macrophages (Fig 5D). Although it has been shown before that SseF and SseG can be shared among *Salmonella* residing in the same cell [40], we show here that all SPI-2 T3SS effectors necessary for survival and proliferation of *Salmonella* ΔSPI-2 can be provided *in trans* at levels sufficient for complementation of the SPI-2 deletion. This suggests that the SPI-2 ON bacteria facilitate the survival and proliferation of SPI-2 OFF bacteria.

## SPI-2 OFF *Salmonella* provide the growth pool for SPI-2 ON bacteria

Expression of the SPI-2 T3SS and its effectors likely impose a significant metabolic burden in the nutrient-limited SCV, a burden not experienced by SPI-2 OFF bacteria protected by SPI-2 ON bacteria in the same host cell. As a result, SPI-2 OFF *Salmonella* may proliferate faster than SPI-2 ON bacteria in the same host cell. To test this hypothesis, we examined individual *Salmonella* WT + pP*sifA*-BFP_P*steD*-GFP for both proliferation of and reporter expression in MgMES pH 5.0 using automated fluorescence microscopy followed by a deep learning analysis algorithm (S6A–S6C Fig).

After an initial lag phase, *Salmonella* WT + pP*sifA*-BFP_P*steD*-GFP proliferated exponentially until the colony size precluded analysis of individual bacteria (S6D Fig). As expected, the median fluorescence intensity of all bacteria (S6E Fig) increased over time in MgMES pH 5.0 during the experiment. However, only subset of *Salmonella* WT + pP*sifA*-BFP_P*steD*-GFP exhibited significant increase in BFP and GFP fluorescence, while the majority showed only minor changes in fluorescence intensity. The percentage of fluorescent bacteria peaked around 2 h after transfer to MgMES pH 5.0, then declined throughout the rest of the experiment (S6F Fig). Interestingly, this decline correlated with the onset of bacterial proliferation after the lag phase, suggesting that non-fluorescent bacteria with inactive *sifA* and *steD* promoters had either shorter lag times or proliferated more rapidly.

To analyse the rate of *Salmonella* WT + pP*sifA*-BFP_P*steD*-GFP proliferation, we analysed distribution of individual bacteria into separate populations based on BFP and GFP fluorescence using automated unbiased FlowSOM algorithm. This analysis identified 3 bacterial subpopulations (S6G Fig), with higher BFP and GFP fluorescence correlating with longer doubling times (S6H Fig).

During the experiment, some initially BFP$^-$GFP$^-$ bacteria began to show BFP and GFP fluorescence (S6A Fig). This further supports our finding that SPI-2 T3SS expressional pattern is not heritable (Fig 3) and can change even during a single infection event. These results suggest that SPI-2$^{OFF}$ *Salmonella* may eventually give rise to SPI-2$^{ON}$ *Salmonella*, potentially replenishing the effector-expressing subpopulation.

To determine whether the lower reporter fluorescence in faster proliferating bacteria reflected lower effector expression or resulted from fluorescence dilution during consecutive cell divisions, we measured fluorescence dilution of GFP pre-synthesised from an inducible promoter using flow cytometry (S7A and S7B Fig). Growth of *Salmonella* WT + pP*ssaG*-BFP_P*BAD*-GFP to stationary phase in the presence of arabinose, which induces P*BAD*, led to a uniform GFP expression (S7C Fig). As expected, subsequent subculture into arabinose-free MgMES pH 5.0 induced P*ssaG* activity in part of the bacterial population, while the amount of pre-synthesised GFP decreased with bacterial proliferation (S7D and S7E Fig). Importantly, the SPI-2$^{ON}$ bacteria showed higher GFP fluorescence than the SPI-2$^{OFF}$ bacteria when the culture reached stationary phase in MgMES pH 5.0 at 8 h post subculturing (S7F Fig). In contrast, we detected no fluorescence dilution when GFP was expressed constitutively from chromosome (*Salmonella glmS::*P*trc*-GFP, S7D Fig) throughout the 8 h growth period in MgMES pH 5.0 mimicking the SPI-2 inducing environment of SCV, suggesting that expression from SPI-2 promoters and maturation time of GFP is short enough to limit the fluorescence dilution effect when its promoter is active in the nutrient-limited conditions.

To investigate proliferation rates during infection, we tracked dilution of GFP synthesised by bacteria prior to host cell infection using flow cytometry. We infected RAW264.7 and Mel JuSo cells with *Salmonella* WT + pP*ssaG*-BFP_P*BAD*-GFP grown in the presence of arabinose, which induces P*BAD*, leading to homogenous GFP expression (Fig 6A). Infection of RAW264.7 macrophages for 6 h in the absence of arabinose allowing for dilution of pre-synthesized GFP between individual daughter cells, was sufficient for detection of an emerging SPI-2$^{ON}$ *Salmonella* subpopulation (Fig 6B). Although GFP fluorescence of SPI-2$^{ON}$ bacteria was decreased compared to pre-infection levels, it remained significantly higher than in GFP$^-$BFP$^-$ bacteria (Fig 6B panel *ii*). The majority of SPI-2$^{ON}$ bacteria lost their preexisting GFP fluorescence by 8 h p.i. (Fig 6C panel *ii*). However, the strength of SPI-2 expression inversely correlated with bacterial proliferation rates (Fig 6B–6D panels *iii*).

Infection of Mel JuSo cells with *Salmonella* WT + pP*ssaG*-BFP_P*BAD*-GFP for 24 h resulted in the formation of two distinct SPI-2$^{ON}$ subpopulations (Fig 6D). While GFP$^{Low}$ SPI-2$^{ON}$ bacteria showed GFP dilution similar to that of GFP$^-$ SPI-2$^{OFF}$ bacteria, the GFP$^{High}$ SPI-2$^{ON}$ bacteria proliferated at a moderate rate (Fig 6D). Interestingly, a subset of SPI-2$^{OFF}$ bacteria maintained high GFP fluorescence throughout the experiments (Fig 6B–6D).

To determine the origin of the non-proliferating GFP$^{High}$ SPI-2$^{OFF}$ population and test whether these bacteria could be intramacrophage-induced persisters [4], we examined SPI-2 T3SS effector expression in non-proliferating intracellular bacteria. Consistently with previous findings [41], we detected effector expression in persisters (S8A Fig). However, significantly fewer effector-expressing bacteria were found among persisters compared to actively growing intracellular bacteria (S8C Fig). The effector-expressing persisters appeared to split further into low-expressing and high-expressing subpopulations (S8A and S8B Fig). These results suggest that the observed GFP$^{High}$ SPI-2$^{OFF}$ subpopulation (Fig 6) consists of persisters and bacteria killed after their entry into the host cells.

If SPI-2$^{OFF}$ *Salmonella* proliferate faster, their relative proportion among intracellular bacteria should increase in time. To test this hypothesis, we reanalysed all our experiments examining the proportion of non-fluorescent bacteria at 8, 10 and 16 h p.i. of RAW264.7 macrophages. Indeed, we found a significant increase in the proportion of SPI-2$^{OFF}$ bacteria at later infection stages (Fig 6E). Collectively, our results suggest that the SPI-2$^{ON}$ bacteria pay a metabolic cost related to effector expression, leading to slower proliferation compared to SPI-2$^{OFF}$ bacteria.

## SPI-2$^{OFF}$ bacteria drives *Salmonella* escape from infected host cells

The intracellular proliferation of *Salmonella* in host cells eventually leads to host cell death and the escape of *Salmonella* into the extracellular environment [33,42]. To examine whether the faster-proliferating SPI-2$^{OFF}$ *Salmonella* escapes more

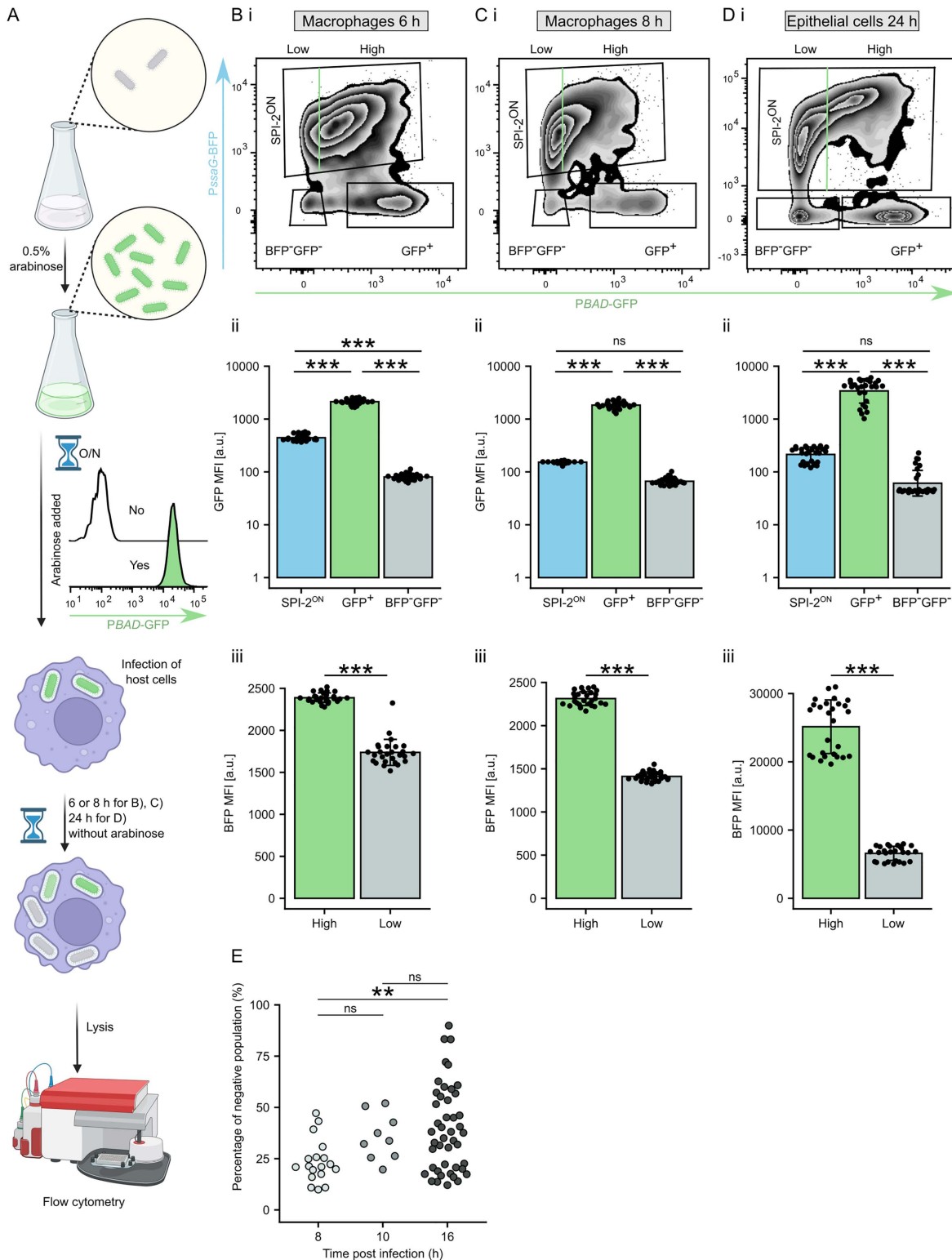

**Fig 6. SPI-2^ON *Salmonella* replicates slower in host cells.** (A) Schematic of GFP fluorescence dilution in *S*Tm + pP*ssaG*-BFP_P*BAD*-GFP upon infection of host cells. *S*Tm + pP*ssaG*-BFP_P*BAD*-GFP was cultured in presence of 0.5% arabinose to induce uniform GFP expression (inset) until the infection of RAW264.7 macrophages (*B* and *C*) or Mel JuSo epithelial cells (*D*) that was in the absence of arabinose. The infected host cells were lysed

at indicated time p.i. and BFP and GFP fluorescence was examined in bacteria from cell lysate by flow cytometry. Loss of GFP fluorescence indicates high replicative capacity. Created in BioRender. Pospíšilová, M. (2025) https://BioRender.com/4ble4tj. (B and C) RAW264.7 macrophages were infected as described in *A)* for 6 h (*B*) or 8 h (*C*). (*i*) Representative zebra plot showing GFP dilution in the SPI-2ON and SPI-2OFF bacteria shows 'SPI-2ON' (BFP+), BFP-GFP- and GFP+ populations. (*ii*) Median of GPF fluorescence of 'SPI-2ON', BFP-GFP- and GFP+ populations. ns, p > 0.05; \*\*\*p < 0.001 (One-way ANOVA with Tukey HSD test). (*iii*) The 'SPI-2ON' population was divided into 'GFPHigh' and 'GFPLow' subpopulations and medians of their BFP fluorescence intensity was compared. Data are from 3 independent experiments in technical triplicates and show means ± SD. \*\*\*p < 0.001 (Paired two-sample t-test). (D) Mel JuSo cells were infected as described in *A)* for 24 h. (*i*) Representative zebra plot showing GFP dilution in the SPI-2ON and SPI-2OFF bacteria shows 'SPI-2ON' (BFP+), BFP-GFP- and GFP+ populations. (*ii*) Median of GPF fluorescence of 'SPI-2ON', BFP-GFP- and GFP+ populations. ns, p > 0.05; \*\*\*p < 0.001 (One-way ANOVA with Tukey HSD test). (*iii*) The 'SPI-2ON' population was divided into 'GFPHigh' and 'GFPLow' subpopulations and medians of their BFP fluorescence intensity was compared. Data are from 3 independent experiments in technical triplicates and show means ± SD. \*\*\*p < 0.001 (Paired two-sample t-test). (E) Increase of the SPI-2OFF *Salmonella* population in time. Reanalysis of all experiments examining the percentage of the SPI-2OFF population (BFP-GFP-) in *S*Tm WT + pP*sifA*-BFP_P*steD*-GFP in 8 h, 10 h, and 16 h p.i. Each dot represents percentage of SPI-2OFF (BFP-GFP-) bacteria from a single independent experiment. \*\*p < 0.01 (Two-sample t-test with Welch's correction).

readily from infected cells, we infected RAW264.7 macrophages with *Salmonella* WT + pP*ssaG*-GFP until the bacteria start to escape from host cells (Fig 7A) probably due to the onset of host cell death [43]. A comparison of bacteria escaped into the extracellular space and those remaining in still-living host cells revealed a significantly higher proportion of the SPI-2OFF bacteria in the extracellular space than within living host cells (Fig 7B), confirming that the faster growing *Salmonella* population is the major cause of host cell death. Similarly, the majority of bacteria still residing in live RAW264.7 or Mel JuSo cells exhibited high activity of P*sifA* and P*steD*, while only small fraction of extracellular bacteria was fluorescent (Fig 7C and 7D) suggesting that bacteria escaping into the extracellular space originated from the rapidly proliferating SPI-2OFF population.

Collectively, our results have revealed that the expression of the SPI-2 T3SS and its effectors is bimodal. This bimodality was regulated on the level of the SPI-2 master regulator two-component system SsrA-SsrB. The SPI-2ON *Salmonella* created a permissive niche in the SCV, enabling SPI-2OFF *Salmonella* inhabiting the same host cell to proliferate, escape from the host cell, and potentially disseminate into new host cells.

## Discussion

The limitation of nutrient availability in pathogen-containing vacuoles is a critical host immune mechanism for controlling the pathogen survival and proliferation. Therefore, adaptation to this evolutionary pressure is crucial for bacterial pathogens with a vacuolar lifestyle. In case of *Salmonella*, several SPI-2 T3SS effectors are involved in such adaptation. Among these SifA, used as an example in this study, is crucial for SCV positioning and membrane dynamics as well as for detoxification of lysosomes thus supporting *Salmonella* survival [44]. Here, we describe in detail the bimodal expression of the SPI-2 T3SS and its effectors. This bimodality likely results from both heterogeneous host cues and the heterogeneous activation of the SPI-2 expression master regulator SsrB. The SPI-2OFF *Salmonella* proliferated more rapidly and egressed in bigger quantities from infected host cells, suggesting that the bimodal expression of the SPI-2 T3SS and its effectors might represent a mechanism distributing the virulence-related energetic burden among all *Salmonella* residing in a single host cell.

Previously, the bimodal expression of other *Salmonella* virulence factors, such as flagella and the SPI-1 T3SS, has been studied using reporter systems analogues to those used here [9,10]. Heterogeneous SPI-2 T3SS expression has also been observed, although its physiological siginificance has not been addressed [18,19,45]. We demonstrate that not only the expression of the SPI-2 T3SS (Fig 1) but also of its effectors (Fig 2) is bimodal both *in vitro* and *in vivo*. We have, however, observed more heterogeneouse effector expression *in vivo* than *in cellulo* (Fig 2E vs 2G). This may be due to different timing of the *in vivo* experiments (i.e., 5 days *in vivo* versus 16 h *in cellulo*), stability of fluorescent proteins over days connected to different kinetic and stability of expression programs, or the numbers of bacteria within individual host cells. A single *Salmonella* is either SPI-2ON or would not proliferate in one host cell. When the one bacterium within a host cell divides and one of the daughter cells turns into the SPI-2OFF state, the fluorescent proteins would be eventually

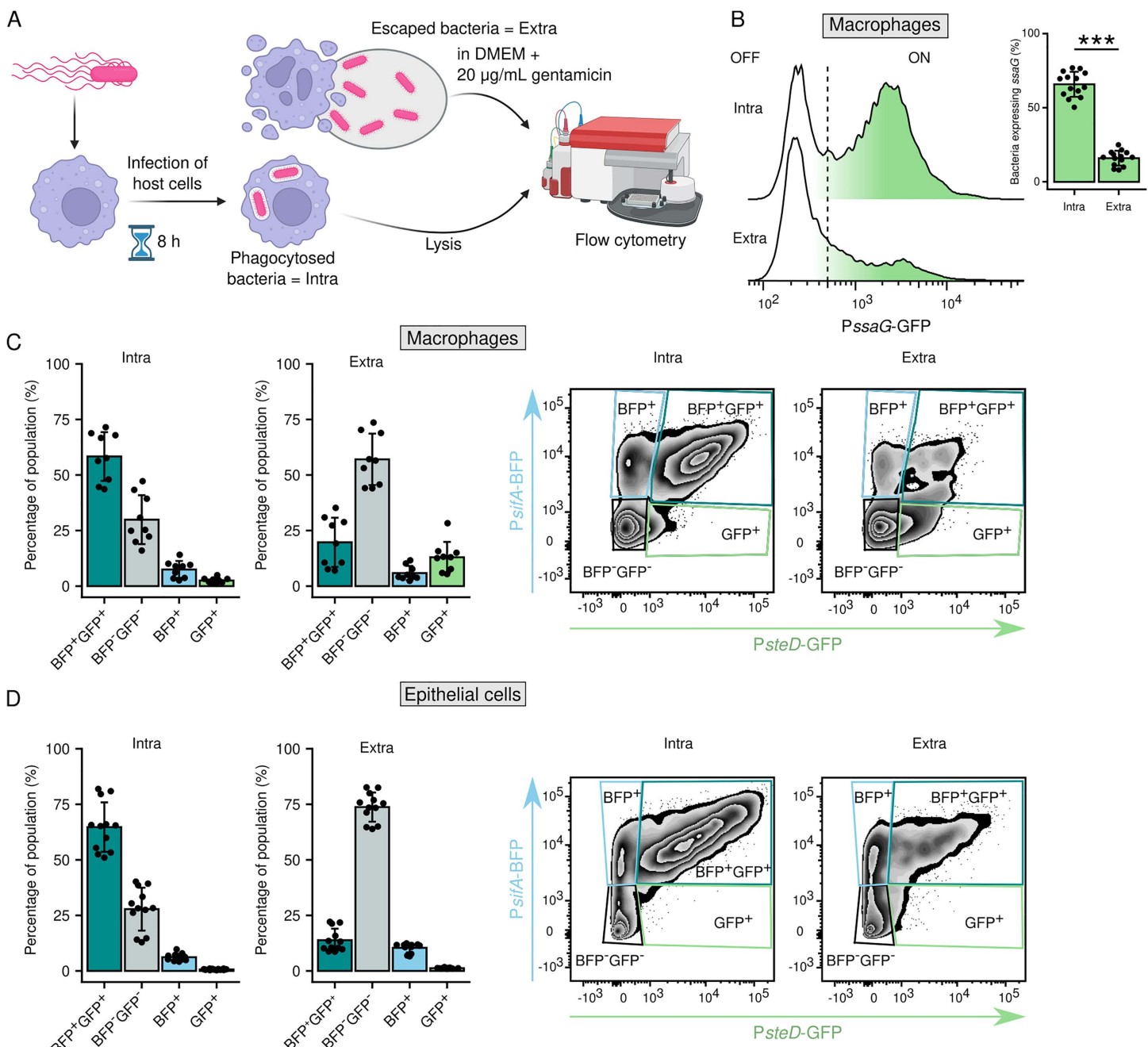

**Fig 7. Salmonella escaping from infected cells is predominantly SPI-2OFF.** (A) Schematic of SPI-2 activity measurement in *Salmonella* escaped from infected cells. Macrophages or epithelial cells were infected with *Salmonella* reporter strains. Extracellular bacteria were killed with gentamicin and washed away 2 h before the end of the experiment. The BFP and GFP fluorescence of bacteria escaped into the extracellular medium (DMEM with 20 µg/mL gentamicin) during the last 2 h of the experiment was measured by flow cytometry. Created in BioRender. Pospíšilová, M. (2025) https://BioRender.com/c97s213. (B) RAW264.7 macrophages were infected for 8 h with *S*Tm WT+pP*ssaG*-GFP. P*ssaG* activity in intracellular bacteria (Intra) from cell lysate and in extracellular bacteria (Extra) collected from supernatant of infected macrophages was monitored as GFP fluorescence by flow cytometry. Data are from 5 independent experiments in technical triplicates and show means±SD. ***p<0.001 (Paired two-sample t-test). (C) RAW264.7 macrophages were infected for 8 h with *S*Tm WT+pP*sifA*-BFP_P*steD*-GFP. P*sifA* and P*steD* activity in intracellular bacteria (Intra) from cell lysate and in extracellular bacteria (Extra) collected from supernatant of infected macrophages was monitored as BFP and GFP fluorescence by flow cytometry. Data are from 3 independent experiments in technical triplicates and show means±SD. (D) Mel JuSo cells were infected for 24 h with *S*Tm WT+pP*sifA*-BFP_P*steD*-GFP. P*sifA* and P*steD* activity in intracellular bacteria (Intra) from cell lysate and in extracellular bacteria (Extra) collected from supernatant of infected Mel JuSo was monitored as BFP and GFP fluorescence by flow cytometry. Data are from 4 independent experiments in technical triplicates and show means±SD.

degraded. This change of expressional pattern would be observable *in cellulo*, while the length of the experiment *in vivo* would limit the chance of identifying these bacteria. The other daughter cells must still express the SPI-2 T3SS to enable further proliferation. While *Salmonella* can proliferate in high numbers *in cellulo* quickly and thus develop several expressional states among all bacteria at the same time, the host environment *in vivo* is probably more restrictive not allowing high heterogeneity among few bacteria. Indeed, we have observed several different expressional patterns in respect to SPI-2 effectors.

The SPI-2 T3SS effectors appear to cluster into two distinguishable groups (S2 Fig) – those involved in SCV regulation, that are expressed by higher percentage of bacteria *in cellulo*, and those not - suggesting the less-expressed effectors may be subject to an additional layer of regulatory control. In contrast, the SPI-1 T3SS effectors are expressed simultaneously, stored within the bacteria until contact with host cells, and then translocated in a stepwise manner [46]. The distinct expression patterns of effectors related and unrelated to the SCV regulation might help with identification of biological functions of lesser-known effectors in future studies.

We observed slower proliferation of *Salmonella* with higher reporter signals both *in vitro* (S6 Fig) and in infected cells (Fig 6). The expression of the SPI-1 T3SS has also been shown to retard *Salmonella* proliferation *in vitro* [11]. The bimodality of the SPI-1 T3SS expression is regulated by the interaction between the repressor HilE and the transcriptional activator HilD, which in turn regulates the activity of the master regulator HilA [10]. In contrast, the regulation of the SPI-2 T3SS expression by the SsrA-SsrB two-component system is more direct. We show that SPI-2 T3SS bimodality is likely regulated at two distinct levels. First, our *in vitro* experiments revealed heterogenous expression of the SPI-2 T3SS and its effectors under uniform conditions (Figs 1B–1E, 2B and S1), suggesting that bimodality is, to some extent, enabled by the structure of the gene expression regulatory circuit in *Salmonella*. Indeed, our experiments demonstrated that overexpression of the active positive regulator SsrB$^{D56E}$ can override the inherent bimodal setup of the SPI-2 expression, indicating that bimodality is regulated on the level of SsrB activation by SsrA (Fig 4).

However, we and others have observed a higher percentage of SPI-2$^{OFF}$ bacteria following infection of both phagocytic as well as non-phagocytic cells in comparison to bacteria grown in minimal media (Fig 1D, 1H and 1I; 19). Since each *Salmonella* resides in a separate yet interconnected SCV [47], the environmental conditions triggering SPI-2 expression may differ slightly around individual bacteria representing external clues as the second regulation level. Consistent with this, the pH of individual phagosomes in macrophages was shown to vary [48,49]. Furthermore, individual *Salmonella* respond heterogeneously to various stressors and limitations within RAW264.7 macrophages [50,51]. Thus, heterogeneity in host signals inducing SPI-2 expression represents another regulatory level of SPI-2 T3SS expression bimodality.

Although the basic regulatory pathway of the SPI-2 expression was described long ago, details of the molecular mechanisms governing its activation and fine-tuning are still being uncovered. It has been shown that SsrB can be phosphorylated in the absence of SsrA by other phosphorylated molecules *in vitro* [11]. Moreover SsrA-independent activation of gene expression by SsrB has been previously described [52]. This may partly explain our results showing that homogeneous overexpression of SsrB leads to increased and unimodal activation of the *ssaG* promoter in the absence of *ssrA* (Fig 4A and 4B). However, the presence of *ssrA* allows significantly stronger SPI-2 T3SS expression, though only in a subset of the population (Fig 4C). The influence of SsrA on SsrB is commonly studied using the SsrB$^{D56E}$ phosphomimetic activatory mutation [34], although this mutant was shown to be unable to bind DNA, with its activity attributed to an overexpression artifact [15]. However, in our experiments, SsrB$^{D56E}$ was able to increase P*ssaG* activity more than equivalently expressed SsrB$^{WT}$ in the absence of SsrA (Fig 4B). Furthermore, unlike the SsrB$^{WT}$, the SsrB$^{D56E}$ mutant induced homogeneous P*ssaG* activity in the presence of SsrA (Fig 4C). Therefore, our results cannot be attributed to an overexpression artifact. In addition, lower P*ssaG* activity in the SsrB$^{D56E}$ mutant than in the SPI-2$^{ON}$ population of either WT or the *ssrB* mutant complemented with *ssrB*$^{WT}$ (Fig 4C) suggests that phosphorylation of D56 is necessary but perhaps not sufficient for full SsrB activation or that the D56E mutation is not truly phosphomimetic. This leads to a conclusion that

SsrA-dependent activation of SsrB drives the bimodal expression of the SPI-2 T3SS. So far, we have been unable to identify genes other than *ssrAB* that drive unimodal SPI-2 T3SS expression.

Bimodal expression gives rise to phenotypic heterogeneity, which is often crucial for the survival and spread of bacterial pathogens. The emergence of *Salmonella* persisters, a subpopulation of bacteria tolerant to antibiotics due to transient growth arrest, is one such example. *Salmonella* persisters express and translocate SPI-2 T3SS effectors, thereby influencing host cells [41]. Here, we show that among the persisters a subpopulation of bacteria does not express SifA and SteD effectors, similar to the pattern observed among growing *Salmonella* (S8 Fig). This suggests that antibiotic persistence and the SPI-2 T3SS bimodal expression are distinct phenomena regulated independently. This may contrast with the proposed interplay between SPI-1 T3SS bimodality and persistence [53]. The bimodal expression of the SPI-1 T3SS is essential for the evolutionary stability of virulence in *Salmonella* [7]. The SPI-1[OFF] and SPI-1[ON] bacterial subpopulations cooperate during host cell invasion [9]. Here, we demonstrate cooperation between bacteria residing in the same host cells yet separated with the SCV membrane. SPI-2[ON] *Salmonella* provide the SPI-2 T3SS effectors enabling efficient proliferation of the SPI-2[OFF] bacteria in the nutrient-limited intravacuolar environment (Fig 5). SPI-2[OFF] *Salmonella* then proliferate more rapidly (Fig 6) and also egress from the infected host cells more efficiently (Fig 7). Upon entering new host cell, SPI-2[OFF] *Salmonella* re-establish the bimodal expression pattern (Fig 3) and contribute to the pool of SPI-2[ON] bacteria during proliferation (S6 Fig). This highlights the plasticity of expression of the SPI-2 T3SS and its effectors and suggests that SPI-2[OFF] *Salmonella* might spread to new host cells more readily.

Other pathogenic bacteria also exhibit bimodal behaviour, thereby creating specialised subpopulations [54–57]. However, very little is known about the interactions between these subpopulations and their physiological impact on the host. Investigating the heterogeneous behaviour of *Salmonella* within individual host cells may provide further insights into this phenomenon in the future.

## Materials and methods

### Ethics statement

Experiments involving infection of C57BL/6 mice were conducted in accordance with the Animal Welfare Committee of the Institute of Molecular Genetics of the Czech Academy of Sciences, v. v. i., in Prague, Czech Republic. Handling of animals was performed according to the Guidelines for the Care and Use of Laboratory Animals, the Act of the Czech National Assembly, Collection of Laws no. 246/1992. Permission no. 54–2022-P were issued by the Animal Welfare Committee of the Institute of Molecular Genetics of the Czech Academy of Sciences in Prague.

### Bacterial strains

Bacteria were grown in Luria–Bertani (LB) medium supplemented with ampicillin (100 µg/mL), kanamycin (100 µg/mL) or chloramphenicol (30 µg/mL) as appropriate. See S1 table for all bacterial strains.

### Plasmid construction

All primers used are listed in S2 Table. Promoter-reporter fusions were created using overlap-PCR [58] and inserted into the pFCcGi plasmid [59] using either BamHI-NotI (for mTagBFP2 reporter) or NsiI-SphI (for GFPmut3 reporter). Promoter regions P*sifA* [60], P*steD* [61], P*ssaG* [62], P*sseJ* [63], P*steC* [64], P*sifB* [60] and P*sseL* [65] were adapted from previously conducted studies. The *ssrB* and *ssrB*[D56E] genes were inserted into the pWSK29 plasmid [66] using HindIII-BamHI. The *ssrB*[D56E] mutation was created using overlap-PCR [58]. To insert mCherry into the Tn7 loci in *Salmonella* genome, P*trc*-mCherry was inserted into pGRG25 as previously described using NotI-SphI and subsequently integrated downstream *glmS* on chromosomal DNA [67]. All plasmids were checked by Sanger sequencing.

## Cell culture

For experiments throughout the article, RAW264.7 and Mel JuSo were selected as model host cells. While RAW264.7 macrophages are routinely used for *Salmonella* research, Mel JuSo represent epithelial cells that do not support *Salmonella* hyperproliferation in the cytosol. Mel JuSo have been used for *Salmonella* research by us and others previously [68]. RAW264.7 mouse macrophages and human Mel JuSo cells were maintained in Dulbecco's modified eagle medium (DMEM; Sigma) supplemented with 10% heat-inactivated endotoxin-free fetal calf serum (FCS; Sigma).

## Cell infection

For infection of RAW264.7 macrophages, overnight Luria broth (LB) cultures of *S.* Typhimurium strains (OD$_{600}$~3.5 to 4.0) were added to adherent macrophages at an MOI of 6:1, centrifuged at 110 *g* for 5 min and incubated at 37°C for 30 min. Cells were washed 3 times with PBS and incubated in fresh medium containing gentamicin (100 µg/mL) to kill extracellular bacteria. After 1 h, the antibiotic concentration was reduced to 20 µg/mL, and cells were processed at indicated times post infection (p.i.).

For infection of Mel JuSo cells, overnight LB cultures of *S.* Typhimurium strains were subcultured 1:100 into fresh LB and incubated with shaking at 37°C to late exponential phase (OD$_{600}$ = 1.8) before being added to Mel JuSo cells at an MOI of 100:1 for 30 min. Cells were washed 3 times with PBS and incubated in fresh medium containing gentamicin (100 µg/mL) to kill extracellular bacteria. After 1 h, the antibiotic concentration was reduced to 20 µg/mL, and cells were processed 24 h p.i.

At the end of the experiment, the host cells were washed with PBS and lysed by hypotonic lysis in deionised water. The bacteria were fixed with 4% PFA at 37°C for 10 min prior to flow cytometry measurement analysis of effector expression.

## Mice

6 to 8 week old C57BL/6 mice (Charles River) were housed as 6 mice per individually ventilated cage under Specified Pathogen Free conditions. For infection experiments for detection of effector expression, mice were inoculated by oral gavage with 1x10$^{10}$ colony forming units (CFU) of stationary phase *S.* Typhimurium 14028 + pP*sifA*-BFP_P*steD*-GFP (OD$_{600}$~3.5 to 4.0) in 200 µL of PBS containing 3% NaHCO$_3$. For competitive infection experiments, mice were injected intraperitoneally with mixture of 5x10$^6$ colony forming units (CFU) of stationary phase *S.* Typhimurium 14028 *glmS*::mCherry or *S.* Typhimurium 14028 *glmS*::GFP (either WT or ΔSPI-2::Cm; OD$_{600}$~3.5 to 4.0) in 200 µL of sterile PBS.

Spleens and mesenteric lymph nodes were isolated at indicated time after infection and enzymatically dissociated in the presence of collagenase IV (1 mg/mL) and DNAse I (0.1 mg/mL) in RPMI-1640 supplemented with 10 mM HEPES and 2% FCS for 30 min at 37°C. Enzymatic digestion was stopped with addition of EDTA (2 mM final concentration). Cells were subsequently collected through a 70 µm cell strainer on ice and washed twice with Ca$^{2+}$, Mg$^{2+}$ free HBSS supplemented with 10 mM HEPES and 2% FCS and used for flow cytometry analysis. For analysis of effector expression by flow cytometry, collected cells were pelleted by centrifugation at 500 *g* for 5 minutes and lysed by hypotonic lysis in deionised water. Cell debris was removed by filtration through 35 µm nylon mesh (BD) prior flow cytometry measurement. A fraction of liberated bacteria was plated on nonselective LB agar plates. The colonies were replica-transferred on a selective LB agar to monitor the presence of the reporter plasmid. No plasmid loss was observed throughout the experiments.

## Fluorescence microscopy

To analyse the growth benefit of *Salmonella* using fluorescent microscopy, RAW264.7 macrophages grown on glass coverslips were infected as described above for 8 h. The cells were washed with PBS, fixed in 4% paraformaldehyde in PBS for 20 min at 37°C, labelled with 0.5 µg/mL DAPI for 10 minutes, washed 3 times in PBS and twice in dH$_2$O and then mounted onto glass slides using Aqua-Poly/Mount (Polysciences). Coverslips were imaged using an Olympus IX83 - inverted confocal wide field microscope (Olympus Life Science).

To monitor SPI-2 T3SS expression using the reporters in infected cells by confocal microscopy, Mel JuSo cells grown on glass coverslips were infected as described above for 24 h p.i. The cells were washed with PBS, fixed in 4% para-formaldehyde in PBS for 20 min at 37°C, washed 3 times in PBS and twice in dH$_2$O and then mounted onto glass slides using Aqua-Poly/Mount (Polysciences). Coverslips were imaged using a Stellaris 8 inverted confocal laser-scanning microscope (Leica Microsystems, Germany).

To assess *Salmonella* growth rate and SPI-2 T3SS expression using the reporters *in vitro*, bacteria from stationary culture were washed twice with MgMES pH 5.0 and diluted to 1 x 10$^8$ CFU/mL prior to seeding on a glass coverslip in 2 µL of MgMES pH 5.0. The bacteria were covered with thin agarose pad (1% agarose in MgMES pH 5.0) and grown at 37°C. A microscopy image was taken every 10 minutes to monitor fluorescence of mCherry, GFP, and BFP using a Dragonfly 503 spinning disk confocal microscope (Andor, Oxford Instruments, UK).

### Image analysis

Using Olympus IX83 with cellSens Dimension software images in Z-stack were deconvoluted with Constrained Iterative deconvolution package. Bacteria were detected and counted in infected cells using CellProfiler. Then the data was analysed using the R environment.

Images from Dragonfly spinning disk confocal microscope were analysed using Omnipose deep learning segmentation package (https://omnipose.readthedocs.io/), a project that specializes in bacterial species segmentation. Time-lapse images of mCherry fluorescence were used to identify the objects of interest. The images were further analysed using DeLTA 2.0 (https://delta.readthedocs.io/en/latest/), a deep learning-based image processing pipeline for segmenting and tracking single cells in time. A pretrained tracking model („mothemachine") was used to explore bacterial division-related phenomena, the pre-processing and analysis was preformed using Python 3.11. In total, 50 colonies were analysed.

The obtained tracking results were further processed using custom-written scripts in MATLAB R2023b (The Math-Works, Natick, MA). Fluorescence of GFP and BFP was extracted and lineage trees showing changes in fluorescence intensities were constructed. Fluorescence of mTagBFP2 and GFP of individual events was analysed at the time of their appearance and disappearance (i.e., when a new daughter cell appeared and when it underwent a new division). These values were used by FlowSOM to identify events sharing fluorescent characteristics [69].

### Flow cytometry

To analyse the growth benefit of *Salmonella* using flow cytometry, RAW264.7 macrophages were infected as described above for 8 h, detached from plastic using 5 mM EDTA to limit the mechanical stress, and fixed in 4% paraformaldehyde in PBS for 20 min at 37°C.

The growth benefit was calculated using MFI (median fluorescence intensities) from double-infected macrophages divided by MFI of single-infected macrophages and normalized to identically counted MFI ratio at 0.5 h p. i.

To analyse expression of fluorescent reporters in individual bacteria, the infected host cells were washed with PBS and then lysed by osmotic lysis using dH$_2$O. Liberated bacteria were fixed in 4% paraformaldehyde in PBS for 20 min at 37°C.

Data were acquired using BD LSR II (BD Biosciences) or ZE5 (Biorad) cytometer and analysed using FlowJo software.

### Viability test

For the viability test of the SPI-2$^{OFF}$ *Salmonella,* RAW264.7 and Mel JuSo cells were infected with *Salmonella* WT + pP*ssaG*-BFP_P*BAD*-GFP as described above. 4 h before lysis (*i.e.,* 12 h p.i. for RAW and 20 h p.i. for Mel JuSo), GFP expression was induced by adding 0.5% arabinose. Bacteria were then collected in a lysate and analysed by flow cytometry as described before.

### Fluorescence-activated sorting of bacteria and reinfection

For fluorescence-activated sorting of bacteria from Mel JuSo lysate and subsequent reinfection, Mel JuSo cells were infected as described above. At 24 h p.i. the infected host cells were washed with PBS and then lysed by osmotic lysis using dH$_2$O. Lysate was collected, and bacteria were sorted by BD Influx based on their fluorescence. Sorted bacteria were grown O/N in LB containing ampicillin (100 µg/mL) and the next morning were subcultured 1:100 into fresh LB and incubated with shaking at 37°C to late exponential phase (OD$_{600}$ = 1.8) before being added to a new batch of Mel JuSo cells at an MOI of 100:1. Reinfection then proceeded the same way as the infection. Lysate was collected, and bacteria were measured by BD Influx.

### Test of proliferation rate

To test the metabolic activity of *Salmonella* during infection, *Salmonella* WT + pP*ssaG*-BFP_P*BAD*-GFP was grown over night in LB medium with ampicillin (100 µg/mL) and 0.5% arabinose to induce GFP expression. RAW264.7 and Mel JuSo cells were infected for 6, 8 (RAW264.7 macrophages) or 24 h (Mel JuSo) in absence of arabinose. Bacteria for flow cytometry analysis were collected as described above.

### Quantification and statistical analysis

Statistical significance was calculated using two-way ANOVA, one-way ANOVA followed by Tukey HSD test or One- or two- sample t-test with Bonferroni correction and pairwise comparison if necessary as indicated in figure legends. All statistical analysis was carried out using the R environment version 4.4.0 except for growth benefit from Fig 5 and percentage of infected macrophages from S4 Fig that were carried out in the GraphPad Prism v9 software. Summarising data tables used for statistical analysis are available on Zenodo (https://doi.org/10.5281/zenodo.17199173).

### Test of bimodality

Data were first downloaded in *.csv files from a flow cytometry analysis using the ViolinPlots plugin. Then, GFP fluorescence intensity from each bacteria was tested as a whole for uni- and bimodality in the R environment using the LaplacesDemon package [21].

### Supporting information

**S1 Fig. Expression of SPI-2 T3SS and its effectors is bimodal.** (A) P*ssaG* activity over time. (*i*) GFP expression in *S*Tm WT + pP*ssaG*-GFP was analysed by flow cytometry at indicated time after subculture from late exponential phase (LB, OD 1.8) into MgMES pH 5. (*ii*) Quantification of GFP$^+$ (ON) and GFP$^-$ (OFF) bacteria as represented in (*i*). Data are from 3 independent experiments in technical triplicates and show means ± SD. (B and C) Bimodality of P*ssaG* activity in macrophages (*B*) and epithelial cells (*C*) is not due to general expressional difference. Data from Figure 1*H* and *I* were reanalysed to show GFP/mCherry MFI ratio. Constitutively expressed mCherry from P*trc* was used as a *Salmonella* marker and for normalization to overall transcriptional activity. *S*Tm WT not carrying the pP*ssaG*-GFP were used as a negative control. (D and E) Presence of the reporter system does not represent major burden for intracellular bacteria in RAW264.7 macrophages (*D*) and Mel JuSo epithelial cells (*E*). Representative flow cytometry data from 3 independent experiments comparing bacterial amounts of *S*Tm WT + pP*ssaG*-GFP and *S*Tm WT. (F) Bimodality of P*sifA* and P*steD* activity in macrophages is not due to general expressional difference. Data from Fig 2E were separated to BFP$^+$ (BFP$^+$ with BFP$^+$GFP$^+$ populations) and BFP$^-$ (BFP$^-$GFP$^-$ and GFP$^+$ populations). Constitutively expressed mCherry from P*trc* was used as a *Salmonella* marker and for normalization to overall transcriptional activity. (G) P*sifA* activity over time. (*i*) BFP expression in *S*Tm WT + pP*sifA*-BFP_P*steD*-GFP was analysed by flow cytometry at indicated time after subculture from late exponential phase (OD 1.8 in LB) into MgMES pH 5. (*ii*) Quantification of BFP$^+$ (ON) and BFP$^-$ (OFF) bacteria

as represented in (*i*). Data are from 3 independent experiments in technical triplicates and show means ± SD. (H) P*steD* activity over time. (*i*) GFP expression in *S*Tm WT + pP*sifA*-BFP_P*steD*-GFP was analysed by flow cytometry at indicated time after subculture from late exponential phase (OD 1.8 in LB) into MgMES pH 5. (*ii*) Quantification of GFP⁺ (ON) and GFP⁻ (OFF) bacteria as represented in (*i*). Data are from 3 independent experiments in technical triplicates and show means ± SD. (I) P*sifA* activity over time. (*i*) BFP expression in *S*Tm WT + pP*sifA*-BFP_P*steD*-GFP was analysed by flow cytometry at indicated time after subculture from late stationary phase (16 h in LB) into MgMES pH 5. (*ii*) Quantification of BFP⁺ (ON) and BFP⁻ (OFF) bacteria as represented in (*i*). Data are from 3 independent experiments in technical triplicates and show means ± SD. (J) P*steD* activity over time. (*i*) GFP expression in *S*Tm WT + pP*sifA*-BFP_P*steD*-GFP was analysed by flow cytometry at indicated time after subculture from late stationary phase (16 h in LB) into MgMES pH 5. (*ii*) Quantification of GFP⁺ (ON) and GFP⁻ (OFF) bacteria as represented in (*i*). Data are from 3 independent experiments in technical triplicates and show means ± SD.
(TIFF)

**S2 Fig. Expression of individual SPI-2 T3SS effectors.** (A) Bimodality of P*sifA* activity measured using destabilized reporter in epithelial cells. Mel JuSo cells were infected with *S*Tm WT + pP*sifA*-GFP, or *S*Tm WT + pP*sifA*-GFP-ASV for 24 h before hypotonic lysis. P*sifA* activity was monitored as GFP fluorescence in bacteria from cell lysate by flow cytometry. The bar plot represents quantification of bacteria in depicted gates. Data are from at least 3 independent experiments in at least technical duplicates and show means ± SD. (B-I) Activity of P*sifA*, P*steD*, P*sseJ*, P*steC*, P*gtgE* and P*sifB* in epithelial cells. Mel JuSo cells were infected for 24 h before hypotonic lysis. Promoter activity was monitored as BFP and GFP fluorescence, respectively, in bacteria from cell lysate by flow cytometry. Representative zebra plots (B-F) of *S*Tm WT + pP*sifA*-BFP_P*steD*-GFP (*B*), *S*Tm WT + pP*sseJ*-BFP_P*steD*-GFP (*C*), *S*Tm WT + pP*sifA*-BFP_P*sseJ*-GFP (*D*), *S*Tm WT + pP*steC*-BFP_P*sifA*-GFP (*E*) and *S*Tm WT + pP*steC*-BFP_P*steD*-GFP (*F*) after infection. (G) Bimodality of P*gtgE*, P*steC*, P*sseJ*, P*sifB*, Ps*seL*, P*steD*, P*sifA* activity in epithelial cells. Mel JuSo cells were infected with *S*Tm WT + pP*effector*-BFP_P*steD*-GFP (where the native promoter for *gtgE*, *steC*, *sseJ*, *sifB*, s*seL*, or *sifA* controlled BFP expression respectively) for 24 h before hypotonic lysis. Promoter activity was monitored as BFP and GFP fluorescence, respectively, in bacteria from cell lysate by flow cytometry. The histograms show BFP fluorescence of individual promoters normalized to GFP fluorescence from the same reporter to allow direct comparison between individual reporters. (H, I) Summarising box and whiskers plot of percentages of BFP⁺ (G) and BFP⁺GFP⁺ (H) populations. Data from 2-4 independent experiments in technical mono- to triplicates and show medians, Q1 and Q3. *p < 0.05; **p < 0.01; ***p < 0.001 (One-way ANOVA with Tukey HSD test).
(TIFF)

**S3 Fig. Metabolic activity and a presence in SCV of intracellular *Salmonella*.** (A) SPI-2ᴼᶠᶠ *Salmonella* are metabolically active. (*i*) Schematic of assessment of metabolic activity of SPI-2ᴼᶠᶠ *Salmonella*. RAW264.7 macrophages were infected with *S*Tm WT + pP*ssaG*-BFP_P*BAD*-GFP for 12 h. Subsequently, arabinose was added to the cell media to assess the capacity of intracellular bacteria to metabolize arabinose. P*ssaG* and P*BAD* activity was measured as BFP and GFP fluorescence, respectively, by flow cytometry 4 h after arabinose addition. (*ii*) Representative flow cytometry data for P*ssaG* and P*BAD* activity in individual bacteria residing in RAW264.7 macrophages treated as described in (*i*). (*iii*) The bar charts show ratios of percentages from populations BFP⁺GFP⁺ and BFP⁺, or GFP⁺ and BFP⁻GFP⁻. Data are from 3 independent experiments in technical triplicates and show means ± SD. ***p < 0.001 (Paired two-sample t-test). Created in BioRender. Pospíšilová, M. (2025) https://BioRender.com/ex2ztjy. (B) SPI-2ᴼᶠᶠ *Salmonella* is in SCV. Representative confocal fluorescence microscopy image of GFP expression in *S*Tm WT + pP*ssaG*-GFP in Mel JuSo cells expressing LAMP1-mTurqoise 24 h p.i. The images show a single Z axis layer and X-Z and Y-Z projections (at the bottom and side of the main images, respectively) to show the signal of the SCV marker LAMP1 adjacent to SPI-2ᴼᶠᶠ *Salmonella*. The arrowheads indicate SPI-2ᴼᶠᶠ (GFP⁻) bacteria surrounded with LAMP1. Scale bar, 10 µm in the overview image and 2 µm in the magnified inset images.
(TIFF)

**S4 Fig. SsrB regulation of bimodality.** (A) The amount of SsrB$^{D56E}$-2HA expressed from P$trc$ is constant throughout the growth of *Salmonella* culture. Total levels of SsrB$^{D56E}$-2HA in *S*Tm WT+pP$trc$-$ssrB^{D56E}$-2HA+pP$ssaG$-GFP (SsrB$^{D56E}$) cultures used in Fig 4A was examined for the presence of HA tag using immunoblotting at indicated OD$_{600}$ after subculture from a late stationary phase (16 h in LB) into fresh LB medium and measured at defined OD. *S*Tm WT+pP$ssaG$-GFP (WT) was used as a negative control and anti-DnaK (*Salmonella*) antibody was used for loading control. Representative of 3 independent experiments and quantification of HA signal relative to DnaK signal intensities from 3 experiments represented is shown. Data show means±SD. (B) P$ssaG$ activity is constant in the presence of SsrB$^{D56E}$-2HA. GFP expression in *S*Tm WT+pP$ssaG$-GFP (WT) and *S*Tm WT+pP$trc$-$ssrB^{D56E}$-2HA+pP$ssaG$-GFP (SsrB$^{D56E}$) was analysed by flow cytometry in samples from *A*) at indicated OD$_{600}$. Data are from 3 independent experiments in technical triplicates and show means±SD. (C) The amount of SsrB-2HA and SsrB$^{D56E}$-2HA is comparable during infection. RAW264.7 macrophages were infected with *S*Tm Δ$ssrAB$+pP$trc$-$ssrB$-2HA+pP$ssaG$-GFP or *S*Tm Δ$ssrAB$+pP$trc$-$ssrB^{D56E}$-2HA+pP$ssaG$-GFP for 16 h. Presence of HA tag in intracellular bacteria was examined using immunoblotting and anti-DnaK (*Salmonella*) antibody was used for loading control. Representative of 3 independent experiments is shown.
(TIFF)

**S5 Fig. SPI-2$^{OFF}$ bacteria benefit from activity of effectors translocated by SPI-2$^{ON}$ bacteria in macrophages.** (A) Gating strategy for the detection of macrophages in spleen homogenates from C57BL/6 mice infected by *S*Tm $glmS$::mCherry WT and *S*Tm $glmS$::GFP ΔSPI-2 using flow cytometry. (B) *S*Tm $glmS$::GFP ΔSPI-2 has a significant survival defect in macrophages *in vivo*. Percentage of C57BL/6 splenic macrophages infected with by *S*Tm $glmS$::mCherry WT or *S*Tm $glmS$::GFP ΔSPI-2. **$p < 0.01$ (Paired two-sample t-test). (C) Representative dot plot of growth benefit data shown in Fig 5B. GFP and mCherry fluorescence intensities in splenic macrophages isolated from C57BL/6 mice infected with *S*Tm $glmS$::mCherry WT and *S*Tm $glmS$::GFP ΔSPI-2. The + signs represent median of fluorescence intensity in individual gates. (D) Median fluorescence intensities (MFI) of populations depicted in *C* and Fig 5B. (E) Growth benefit of bacteria from co-infections of RAW264.7 macrophages. Data are calculated from Fig 5Cii as a ratio of double-infected (dark magenta and green) and single-infected cells (light magenta and green). Calculation was done on data from both 8 h p.i. and 0.5 h p.i. (F) *S*Tm $glmS$::GFP ΔSPI-2 has a significant growth defect in comparison to *S*Tm $glmS$::mCherry WT in macrophages. (*i*) Representative histograms showing bacterial burden in RAW264.7 macrophages from Fig 5D at 0.5 h and 8 h p.i. (*ii*) Fold replication values were calculated from the medians of fluorescent intensities of mCherry and GFP fluorescence respectively as the ratio of replication at 8 h to replication at 0.5 h. Data are from 3 independent experiments in technical triplicates and show means±SD. *$p < 0.05$ (Paired two-sample t-test).
(TIFF)

**S6 Fig. SPI-2$^{OFF}$ *Salmonella* has shorter division time than SPI-2$^{ON}$ *Salmonella*.** (A) Time-lapse microscopy monitoring growth of SPI-2$^{ON}$ *Salmonella*. Individual *S*Tm WT+pP$sifA$-BFP_P$steD$-GFP were grown in MgMES pH 5 on agarose pads for 8 h. Fluorescence of BFP and GFP was monitored at 10 min intervals by fluorescence microscopy. (B and C) Lineage tree showing activity of P$sifA$ (B) and P$steD$ (C) in the highlighted bacterial colony in *A*). Colouring of the lineage trees reflects the relative mean BFP intensity of individual cells scaled to the highest intensity in the tree. (D) The increase of the number of *S*Tm WT+pP$ifA$-BFP_P$steD$-GFP expressing or not BFP and GFP from *A*) over time. Data from all 3 independent experiments were pooled together. (E) Overall activity of P$sifA$ and P$steD$ increases in time. The graph shows median fluorescence intensity (MFI) of BFP and GFP fluorescence of all bacteria per time point of the experiment in *A*). Means±SD of data from all 3 independent experiments are shown. (F) Percentage of fluorescent bacteria from *A*) over time. Means±SD of data from all 3 independent experiments are shown. (G) Dot plot showing fluorescence intensities of individual bacteria before division or at the end of the experiment. The bacteria were automatically gated into 3 populations by the FlowSOM software plugin based on fluorescent intensities. Data are from 3 independent

experiments. (H) Doubling time of individual bacteria in populations shown in *G)*. ***p<0.001 (One-way ANOVA with Tukey HSD test).
(TIFF)

**S7 Fig. GFP and BFP dilution in bacteria during growth in vitro.** (A) Gating strategy for detection of GFP by flow cytometry in *S*Tm *glmS::*P*trc*-GFP (Constitutive) bacteria. (B) Gating strategy for detection of GFP and BFP by flow cytometry in *S*Tm *glmS::*P*trc*-mCherry+pP*ssaG*-BFP_P*BAD*-GFP (Inducible) bacteria. (C) GFP expression from an inducible promoter. *S*Tm+pP*ssaG*-BFP_P*BAD*-GFP was cultured in the presence or absence of 0.5% arabinose to induce uniform GFP expression. (D) *S*Tm *glmS::*P*trc*-GFP (Constitutive) grown in the presence of arabinose or arabinose-induced *S*Tm+pP*ssaG*-BFP_P*BAD*-GFP (Inducible) were subcultured into MgMES pH 5 and GFP fluorescence (solid line, main y axis) and $OD_{600}$ (dashed line, secondary y axis) were measured. at indicated time. GFP signal was normalized to the initial time point (0 h after inoculation) of each experiment. Data are from 3 independent experiments in technical triplicates and show mean±SD. (E) P*ssaG* activity of *S*Tm+pP*ssaG*-BFP_P*BAD*-GFP (Inducible) from *D*). BFP was normalized to the initial time point (0 h after inoculation) of each experiment. Normalized BFP fluorescence (solid line, main y axis) and $OD_{600}$ (dashed line, secondary y axis) are from 3 independent experiments in technical triplicates and show mean±SD. (F) P*BAD* activity in the SPI-2$^{ON}$ and SPI-2$^{OFF}$ populations of pP*ssaG*-BFP_P*BAD*-GFP after 8 h in MgMES pH 5. Data are from 3 independent experiments in technical triplicates and show mean, Q1, Q3 and grey lines connecting replications. ***p<0.001 (Paired two-sample t-test).
(TIFF)

**S8 Fig. Bimodality of P*sifA* and P*steD* activity in persisters in macrophages.** (A) Bimodality of P*sifA* and P*steD* activity in persisters in macrophages. RAW264.7 macrophages were infected with *S*Tm WT+pP*sifA*-BFP_P*steD*-GFP. The extracellular and intracellular proliferating bacteria were killed using cefotaxime. Gentamicin killing only the extracellular bacteria was used as a control. The host cells were lysed by hypotonic lysis 16 h p.i. and P*sifA* and P*steD* activity was monitored as BFP and GFP fluorescence respectively in bacteria from cell lysate by flow cytometry. (B) The SPI-2$^{ON}$ population shows more heterogeneous activity of P*sifA* and P*steD* in persisters. The box-and-whisker plot represents the quantification of Coefficient of Variation (CV) of BFP and GFP fluorescence in bacteria in the gate BFP$^+$GFP$^+$ depicted in *A*. Data are from 3 independent experiments in technical triplicates and show medians, Q1 and Q3. ***p<0.001 (Two-sample t-test). (C) Activity of P*sifA* and P*steD* in persisters differs from growing bacteria. The box-and-whisker plot represents the percentage of bacteria in BFP$^-$GFP$^-$ and BFP$^+$GFP$^+$ gates depicted in *A*. Data are from 3 independent experiments in technical triplicates and show medians, Q1 and Q3. *p<0.05; **p<0.01; ***p<0.001 (Two-sample t-test).
(TIFF)

**S9 Fig. Raw western blotting images used in S4A and S4C Figs. The same antibodies against DnaK (positive control; 69 kDa) and HA tag (on SsrB; together 26 kDa) were used for both figures. In S4A are three repetitions, X represents an HA-negative control.**
(PDF)

**S1 Table. Bacteria strains used in this study.**
(XLSX)

**S2 Table. Primers used in this study.**
(XLSX)

**S3 Table. The values used to build graphs in this study.**
(XLSX)

**S1 File. Raw western blotting images used in S4 Fig.**
(TIFF)

## Acknowledgments

We thank members of the Laboratory of Bacterial Virulence and Laboratory of Infection Biology at the Institute of Microbiology of the Czech Academy of Sciences for helpful suggestions. Mel JuSo and *S.* Typhimurium 14028 were a gift from David Holden (Imperial College London). *glmS::*mCherry scaffold was kindly provided by Leigh Knodler (The University of Vermont). pFCcGi was a gift from Sophie Helaine & David Holden (Addgene plasmid # 59324; http://n2t.net/addgene:59324; RRID:Addgene_59324). We thank Zaneta Slavickova and Jan Svoboda from the Flow cytometry and microscopy core facility of IMIC and Zdenek Cimburek from the Flow cytometry of IMG for help with FACS. The authors used services of the Czech Centre for Phenogenomics at the Institute of Molecular Genetics supported by the Czech Academy of Sciences RVO 68378050 and by the project LM2023036 Czech Centre for Phenogenomics provided by Ministry of Education, Youth and Sports of the Czech Republic. We acknowledge the Light Microscopy Core Facility, IMG, Prague, Czech Republic, supported by grants "National Infrastructure for Biological and Medical Imaging" (MEYS – LM2023050), "Modernization of the national infrastructure for biological and medical imaging Czech-BioImaging" (MEYS – CZ.02.1.01/0.0/0.0/18_046/0016045) and "Modernization of the VVI Czech-BioImaging" (MEYS – CZ.02.01.01/00/23_015/0008205), for their support with the confocal imaging and image analysis presented herein.

## Author contributions

**Conceptualization:** Milada Pospíšilová, Ondrej Cerny.

**Data curation:** Milada Pospíšilová, Ondrej Cerny.

**Formal analysis:** Milada Pospíšilová, Alona Dreus, Paulina Matheova, Michaela Blazikova, Martin Capek, Ondrej Cerny.

**Funding acquisition:** Ondrej Cerny.

**Investigation:** Milada Pospíšilová, Paulina Matheova, Jana Schmidtova, Barbora Pravdova, Ondrej Cerny.

**Methodology:** Milada Pospíšilová, Alona Dreus, Paulina Matheova, Ondrej Cerny.

**Project administration:** Ondrej Cerny.

**Resources:** Milada Pospíšilová, Ondrej Cerny.

**Supervision:** Ondrej Cerny.

**Validation:** Ondrej Cerny.

**Visualization:** Milada Pospíšilová, Alona Dreus, Paulina Matheova, Michaela Blazikova, Martin Capek, Ondrej Cerny.

**Writing – original draft:** Milada Pospíšilová, Ondrej Cerny.

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
