## [Decision Letter · Decision Letter 0]

29 Jul 2025

Bimodal Expression of Type 3 Secretion System 2 Enables Cooperative Virulence among Intracellular Salmonella Typhimurium

PLOS Pathogens

Dear Dr. Cerny,

Thank you for submitting your manuscript to PLOS Pathogens. After careful consideration, we feel that it has merit but does not fully meet PLOS Pathogens's publication criteria as it currently stands. Therefore, we invite you to submit a revised version of the manuscript that addresses the points raised during the review process.

Please submit your revised manuscript within 60 days Sep 27 2025 11:59PM. If you will need more time than this to complete your revisions, please reply to this message or contact the journal office at plospathogens@plos.org. Please include the following items when submitting your revised manuscript:

We look forward to receiving your revised manuscript.

Kind regards,

Nirmal Robinson

Academic Editor

PLOS Pathogens

Thomas Guillard

Section Editor

PLOS Pathogens

Editor-in-Chief

PLOS Pathogens

PLOS Pathogens

orcid.org/0000-0002-7699-2064

**Journal Requirements:**

At this stage, the following Authors/Authors require contributions: Alona Dreus, Paulina Matheova, Jana Schmidtova, Barbora Pravdova, Michaela Blazikova, and Martin Capek. Please ensure that the full contributions of each author are acknowledged in the "Add/Edit/Remove Authors" section of our submission form.

3) Some material included in your submission may be copyrighted. According to PLOSu2019s copyright policy, authors who use figures or other material (e.g., graphics, clipart, maps) from another author or copyright holder must demonstrate or obtain permission to publish this material under the Creative Commons Attribution 4.0 International (CC BY 4.0) License used by PLOS journals. Please closely review the details of PLOSu2019s copyright requirements here: PLOS Licenses and Copyright. If you need to request permissions from a copyright holder, you may use PLOS's Copyright Content Permission form.

Potential Copyright Issues:

i) Figures 3A, 5A. 6A, and 7A. Please confirm whether you drew the images / clip-art within the figure panels by hand. If you did not draw the images, please provide (a) a link to the source of the images or icons and their license / terms of use; or (b) written permission from the copyright holder to publish the images or icons under our CC BY 4.0 license. Alternatively, you may replace the images with open source alternatives. See these open source resources you may use to replace images / clip-art:

4) We note that your Data Availability Statement is currently as follows: "The published article includes all datasets generated or analysed during this study. Detailed datasets are available on Zenodo (DOI: 10.5281/zenodo.15309578).". Please confirm at this time whether or not your submission contains all raw data required to replicate the results of your study. Authors must share the “minimal data set” for their submission. PLOS defines the minimal data set to consist of the data required to replicate all study findings reported in the article, as well as related metadata and methods (https://journals.plos.org/plosone/s/data-availability#loc-minimal-data-set-definition).

**Reviewers' Comments:**

Reviewer's Responses to Questions

**Part I - Summary**

Reviewer #1: In this manuscript by Kambova et al., the authors investigate how bimodal expression of Salmonella pathogenicity island 2 type 3 secretion systems (SPI-2 T3SS) is regulated and its influence on the intracellular survival/replication of Salmonella enterica serovar Typhimurium. In agreement with previous studies, they demonstrate that SPI-2 T3SS expression is bimodal among intracellular bacteria using flow cytometry and microscopy. The authors extend on these previous works to reveal bimodal expression of effectors as well and the ability of SPI-2 T3SS+/effector+ bacteria to support intracellular replication and subsequent escape of SPI-2 T3SS-/effector- bacteria within the same host cell. The writing throughout most of the paper is clear and detailed, with a few areas that could be expanded upon. Additionally, there are a few suggestions, including important controls, that the authors should perform to make their claims stronger.

1) In Fig 1B&C, the authors characterize the functionality of their SsaG GFP reporter systems via microscopy and claim that the “majority of bacteria from both reporter strains showed high GFP fluorescence” (line 90-91). However, based on images presented, it looks like less than half of the bacteria are expressing GFP and not all the non-GFP-expressing bacteria are marked with a pink arrow. Can the percentage of GFP expressing bacteria be quantified from these images or the claim adjusted with arrows highlighting all non-GFP-expressing bacteria? However, it is clear from the flow cytometry done in Figure 1D, that a majority of the bacteria are indeed expressing GFP.

2) As a control, can bimodal expression of SsaG, SifA, and SteD also be seen via qPCR in addition to promoter-linked GFP/BFP expression? It’s important that their reporter system accurately reflects the true expression of these proteins.

3) In Fig 1E, Fig S1A, and Fig S1G-J, the percentage of SsaG-, SifA-, and SteD- bacteria decreases over time up to 5-6 hours in MgM-MES pH 5.0. The percentage of SsaG-, SifA-, and SteD- bacteria (represented by GFP/BFP expression) at the end of the kinetic experiment are quite low. Are these populations sustained and significant? Or are SsaG and effector expression temporal rather than consistently bimodal? The following are possible ways of validating:

a. Extending the kinetic experiment to longer time points would help to confirm that these populations are always present and the decrease in their abundance plateaus such as in Fig S1H and S1J.

b. Placing GFP/BFP expression under a constitutive promoter can act as a positive control whereby there should be no GFP-/BFP- populations and demonstrate these low percentages of SsaG-, SifA-, and SteD- bacteria are significant and not an artifact of flow cytometry gating or using GFP/BFP as a proxy for the expression of these proteins. For example, in Fig 6A, it looks like there is also a small GFP- population even under the pBAD promoter.

4) In Fig 3, do the sorted individual subpopulations after the first infection establish the other populations during axenic regrowth to exponential phase in LB or only after infecting Mel JuSo cells? Flow cytometry of the inoculum immediately after sorting and after regrowth prior to second infection would clarify.

5) In addition, none of the calculations in this paper take into account the half-life of GFP or BFP. How do these effect the interpretation of all their results?

6) In Figure 5B and 5Cii, only a ratio of “Double inf. / Single inf.” is shown. Displaying the raw MFI values in individual subpanels, without the ratio, in singly versus doubly infected macrophages for both WT mCherry and ΔSPI-2 GFP would be helpful to emphasize the claim that SPI-2OFF benefit from the effectors of SPI-2ON. The raw data before calculating the ratio should at least be in the supplemental figures.

7) In Figure 5Dii, showing the raw values would be helpful to assess the heterogeneity per condition and the magnitude of difference since the difference in the median values of the violin plots seem minor.

8) In lines 295-298, more description of how bacterial subpopulations are identified and how doubling time is determined will be beneficial. What features led the algorithm to separate out each subpopulation? Are there enough individual bacteria in Population 3 to form its own population?

9) The captions for Fig S7 need to be fixed. The text does not align with Fig S7.

a. It is not clear how GFP fluorescence and PssaG activity are normalized.

b. Fig S7D is important for the claim that SPI-2ON bacteria replicate slower than SPI-2OFF via GFP dilution. However, it is not clear based on the main text and the captions of the panel how this experiment was done. Fixing the captions and providing more information in the main text (ie time point, whether FACS sorting was involved, etc) will help. It seems like Fig S7D is currently linked with the caption for Fig S7F; if so, how were the quadrants defined? Gating strategy should be shown.

10) For the GFP dilution infection experiments described in Fig 6, is the presence of L-arabinose eliminated prior to infection with macrophages? This should be clarified in the text and Fig 6A schematic as it affects how the data is interpreted.

a. The reason why the four separate populations are defined is not clear in the text. If the goal is to compare the proliferation rates between SPI-2ON and SPI-2OFF, GFP MFI of SsaG+ (BFP+) vs SsaG- (BFP-) would be a more direct comparison instead of comparing SsaG+ (BFP+) separately with GFP+ and GFP- BFP- as shown.

b. To quantify GFP dilution as a proxy for replication rate, the GFP MFI between two time points as a “fold replication” is helpful ie between 2 hrs and 6/8 hrs (Figueira et al. mBio 2013 & Hamblin et al. mBio 2023). Here, the authors are comparing GFP MFI of a single time point without normalizing to where the initial GFP MFI starts at upon SsaG formation.

c. Minor note, it would be more consistent to provide one label instead of switching between T3SS-2ON and SPI-2ON.

11) In Fig 6E, there isn’t statistical significance, so the claim in lines 339-340 of a “significant increase” needs to be changed.

12) In Fig 7, the authors claim that extracellular bacteria have a higher proportion of SPI-2OFF than intracellular bacteria. A key assumption is that upon inducing host cell lysis, the expression of SsaG isn’t downregulated from SPI-2ON bacteria once outside of the nutrient-limited SCV and in the extracellular media. Control experiments are needed ie comparing SsaG expression after transferring bacteria from MgM-MES pH 5 media to host cell media in the time scale of the infection/lysis experiments.

13) In line 426, it’s worth specifying that only SteD and SifA were tested in this work in the context of persisters. Would some persisters also not express other effectors?

14) In the Discussion, perhaps the authors can discuss why SifA+ SteD- or SifA- SteD+ populations present in vitro (Fig 2E) are not present upon in vivo infection in C57BL/6 mice (Fig 2F).

15) Very minor but in line 979, “analyzed” is used while “analysed” was used throughout the rest of the text.

Reviewer #2: The manuscript presents a compelling study on the bimodal expression of SPI-2 T3SS and its effectors in Salmonella Typhimurium, exploring its physiological significance during infection. While the work is interesting and potentially impactful, several critical technical and presentation issues need to be addressed to enhance the clarity, rigor, and reproducibility of the findings.

Reviewer #3: The bacterial Type III Secretion System (T3SS) encoded on the Salmonella pathogenicity island 2 (SPI-2) is a key and highly conserved virulence mechanism in many Salmonella serovars. This system enables the translocation of effector proteins from vacuole-bound intracellular bacteria into the cytosol of host cells. These effector proteins support various aspects of Salmonella pathogenesis, including vacuole integrity, intracellular nutrient acquisition, and immune evasion.

In the present manuscript, the authors report that the expression of the SPI-2 T3SS and its effector proteins follows a bimodal pattern. They present a comprehensive analysis using multiple approaches, including flow cytometry, fluorescence microscopy, and both in vitro and in vivo infection models. Their findings show that this bimodal expression is regulated by the response regulator SsrB and that the two subpopulations SPI-2 ON and SPI-2 OFF mutually support each other. Specifically, SPI-2 ON Salmonella secrete effectors that enable efficient proliferation of SPI-2 OFF bacteria within the nutrient-limited intravacuolar environment. In turn, SPI-2 OFF bacteria re-establish the bimodal expression and re-establish the SPI-2 ON sub-population during intracellular replication.

In summary, this is a nicely written manuscript which provides convincing evidence that Salmonella exhibits bimodal expression of the SPI-2 T3SS, which may constitute an adaptive division of labor strategy that enhances Salmonella fitness and dissemination during infection.

**Part II – Major Issues: Key Experiments Required for Acceptance**

Reviewer #1: (No Response)

Reviewer #2: 1. The authors should clarify whether the constructed fluorescent reporter plasmids are stably maintained during in vitro and in vivo infections. What is the loss rate of the plasmid in host cells or mice? How might plasmid loss affect the quantification and interpretation of SPI-2ON versus SPI-2OFF populations? Additionally, the authors should describe in detail the procedures for recovering bacteria from lysed cells or mouse organs and whether these steps could artificially alter SPI-2 expression.

2. Figure 1F and 1G, given that Salmonella expresses red fluorescence and ssaG+ bacteria express green fluorescence, co-localization should appear yellow. However, no yellow fluorescence is visible in the merged images.

Reviewer #3: 1) A major concern is the absence of degradation tags in the SPI-2 reporters used in this study. As a result, the data likely reflect a relatively static view of the bacterial population. SPI-2 OFF bacteria may still appear fluorescent due to previous reporter signal inherited from a SPI-2 ON parent or even grandparent cell. This lack of temporal resolution could affect the interpretation of the expression dynamics and potentially obscure the true proportions of SPI-2 ON and OFF subpopulations. To resolve this issue, the authors should repeat key reporter assays using destabilized fluorescent proteins to enable more accurate, real-time assessment of SPI-2 expression dynamics and reduce signal carryover from ancestral ON states.

2) Figure 7: the authors propose that SPI-2 OFF bacteria are responsible for driving escape from either macrophages or epithelial cells. However, the experimental readout lacks precision. Bacterial exit from host cells is likely asynchronous, meaning that bacteria which escape early may have had up to two hours to replicate extracellularly. During this time, the SPI-2 reporter signal could be rapidly diluted through cell division, potentially leading to an overestimation of the SPI-2 OFF population. To strengthen their conclusion, the authors could complement their escape assays with the TIMER-Bac system (Claudi et al., 2014) to infer the relative age and division history of escaped bacteria, thereby distinguishing recently egressed SPI-2 OFF cells from those that may have undergone extracellular replication and diluted their reporter signal.

**Part III – Minor Issues: Editorial and Data Presentation Modifications**

Reviewer #1: (No Response)

Reviewer #2: 1. The schematic in Figure 1A is unclear. The two bacterial strains used should be explicitly labeled as: S. Typhimurium 12023 glmS::Ptrc-mCherry + pPssaG-GFP and S. Typhimurium 12023 glmS::PssaG-GFP. In Figure 1B, it is not clear whether the red fluorescent protein is being correctly expressed. In Figure 1D, the peaks must be annotated for clarity, and the figure legends need clarify what is MES.

2. Lines 199 and 336, the authors mention that bacteria are "dead." What experimental evidence supports this claim?

3. The rationale for choosing Mel JuSo cells for the infection experiments should be explained. What are the unique features of this cell line that make it suitable for this study?

4. Why did the authors choose to heterologously express LAMP1-mTurquoise rather than use endogenously expressed lysosomal markers for SCV?

5. It would be clearer if Figure 4ii is presented with strain names on the x-axis and GFP fluorescence intensity on the y-axis, which would improve readability for the audience.

6. The SPI-2OFF peak in Figure 4B is noticeably higher than in Figures 1D. This inconsistency should be explained or discussed.

7. The statistical comparisons in Figures 5B and 5Cii are not well explained. Which groups are being compared? Please clearly describe the comparisons and significance testing used.

8. In Figure 5D, why is the bacterial burden of WT in singly infected cells significantly lower than in co-infected cells? Please clarify.

9. Figures 6B and 6C report data from 6 hours and 8 hours post-infection, whereas Figure 6E presents 8 and 16 h time points. Please explain the rationale behind using different time windows.

10. Why was cefotaxime used to remove extracellular and intracellular bacteria in Figure S8? Is there published evidence supporting the use of this antibiotic for such purposes?

11. In line 371, the sentence stating, "The SPI-2OFF Salmonella proliferated more rapidly and egressed more quickly from infected host cells" is misleading. Since cell lysis releases both SPI-2ON and SPI-2OFF populations, this needs to be more accurately stated.

12. Dynamics of SPI-2ON/OFF Ratios, could the authors provide additional time points showing the dynamic changes in the ratio of SPI-2ON and SPI-2OFF bacteria during infection? Is there a consistent trend of SPI-2OFF enrichment in later stages?

13. In lines 390–391, the authors claim regulation occurs at two levels. Please specify these layers explicitly. From the data, only SsrA-SsrB regulation is apparent. Is there additional evidence to support the second regulatory layer?

14. In the Discussion section, please refer to specific figures when discussing experimental results to guide the reader more effectively.

15. Line 866-867, Please clarify why an OD 1.8 is used to represent the late exponential phase, while a time point of 16 hours is used to represent the late stationary phase.

16. Line 889, Please provide complete strain information.

17. Line 919, (ii) changed to (iii).

18. Salmonella should be italicized throughout the manuscript. Units such as “mL” and “μL” should be standardized and formatted correctly across the text.

19. The manuscript requires improved writing and organization. Some paragraphs contain only a single sentence. Additionally, the references are not formatted according to the journal's guidelines.

Reviewer #3: - Line 396: “we and others have observed a higher percentage of SPI-2 OFF bacteria following infection of both phagocytic as well as non-phagocytic cells” -> should read SPI-2 ON bacteria

- Figure 1H and 1I: What is the explanation for 20% more SPI2-ON bacteria in epithelial cells compared to macrophages? A better control would be a ΔssrAB mutant to show that the reporter plasmid is not expressed under any other influence.

- All figures: Where appropriate, individual data points should be shown to facilitate interpretation of the biological variation.

PLOS authors have the option to publish the peer review history of their article (what does this mean? ). If published, this will include your full peer review and any attached files.

**Do you want your identity to be public for this peer review?** For information about this choice, including consent withdrawal, please see our Privacy Policy .

Reviewer #1: No

Reviewer #2: No

Reviewer #3: No

**Figure resubmission:**
---

## [Decision Letter · Decision Letter 1]

23 Oct 2025

PPATHOGENS-D-25-01347R1

Bimodal Expression of Type 3 Secretion System 2 Enables Cooperative Virulence among Intracellular Salmonella Typhimurium

PLOS Pathogens

Dear Dr. Cerny,

Thank you for submitting your manuscript to PLOS Pathogens. After careful consideration, we feel that it has merit but does not fully meet PLOS Pathogens's publication criteria as it currently stands. Therefore, we invite you to submit a revised version of the manuscript that addresses the points raised during the review process.

Please submit your revised manuscript within 30 days Dec 22 2025 11:59PM. If you will need more time than this to complete your revisions, please reply to this message or contact the journal office at plospathogens@plos.org. Please include the following items when submitting your revised manuscript:

We look forward to receiving your revised manuscript.

Kind regards,

Nirmal Robinson

Academic Editor

PLOS Pathogens

Thomas Guillard

Section Editor

PLOS Pathogens

Sumita Bhaduri-McIntosh

Editor-in-Chief

PLOS Pathogens

orcid.org/0000-0003-2946-9497

Michael Malim

Editor-in-Chief

PLOS Pathogens

orcid.org/0000-0002-7699-2064

**Reviewers' Comments:**

Reviewer's Responses to Questions

**Part I - Summary**

Reviewer #2: The revised manuscript has been improved considerably, and the authors have addressed most of the previous concerns in a thorough manner. I have a few minor issues that still require attention:

1. The first author’s name has been corrected in the main author list; however, it still appears as Kambova, M. in Line 908, Line 944, Line 971, and Line 996. Please update these instances accordingly.

2. Regarding the previous comment: “Please clarify why an OD 1.8 is used to represent the late exponential phase, while a time point of 16 hours is used to represent the late stationary phase,” it would be clearer to use a consistent metric for different growth phases. Either represent all phases by OD values or by time points (hours).

3. In Line 269, please verify the spelling of “cellulo.” Ensure that it is correct in the context of the manuscript.

Reviewer #3: The authors have appropriately addressed my comments and I congratulate them on this nice manuscript.

**Part II – Major Issues: Key Experiments Required for Acceptance**

Reviewer #2: (No Response)

Reviewer #3: NA

**Part III – Minor Issues: Editorial and Data Presentation Modifications**

Reviewer #2: (No Response)

Reviewer #3: NA

PLOS authors have the option to publish the peer review history of their article (what does this mean? ). If published, this will include your full peer review and any attached files.

**Do you want your identity to be public for this peer review?** For information about this choice, including consent withdrawal, please see our Privacy Policy .

Reviewer #2: No

Reviewer #3: No

**Figure resubmission:**
---

## [Editor Report · Decision Letter 2]

14 Nov 2025

Dear Dr. Cerny,

We are pleased to inform you that your manuscript 'Bimodal Expression of Type 3 Secretion System 2 Enables Cooperative Virulence among Intracellular Salmonella Typhimurium' has been provisionally accepted for publication in PLOS Pathogens.

Best regards,

Nirmal Robinson

Academic Editor

PLOS Pathogens

Thomas Guillard

Section Editor

PLOS Pathogens

Sumita Bhaduri-McIntosh

Editor-in-Chief

PLOS Pathogens

orcid.org/0000-0003-2946-9497

Michael Malim

Editor-in-Chief

PLOS Pathogens

orcid.org/0000-0002-7699-2064
---

## [Editor Report · Acceptance letter]

Dear Dr. Cerny,

We are delighted to inform you that your manuscript, " 

Bimodal Expression of Type 3 Secretion System 2 Enables Cooperative Virulence among Intracellular Salmonella Typhimurium," has been formally accepted for publication in PLOS Pathogens.

Best regards,

Sumita Bhaduri-McIntosh

Editor-in-Chief

PLOS Pathogens

orcid.org/0000-0003-2946-9497

Michael Malim

Editor-in-Chief

PLOS Pathogens

orcid.org/0000-0002-7699-2064